# Real-World Safety of CFTR Modulators in the Treatment of Cystic Fibrosis: A Systematic Review

**DOI:** 10.3390/jcm10010023

**Published:** 2020-12-23

**Authors:** Renée V. E. Dagenais, Victoria C. Su, Bradley S. Quon

**Affiliations:** 1Adult Cystic Fibrosis Program, St. Paul’s Hospital, Vancouver, BC V6Z 1Y6, Canada; VSu@providencehealth.bc.ca (V.C.S.); bradley.quon@hli.ubc.ca (B.S.Q.); 2Department of Pharmacy, St. Paul’s Hospital, Vancouver, BC V6Z 1Y6, Canada; 3Department of Medicine, University of British Columbia, Vancouver, BC V6Z 1Y6, Canada; 4Centre for Heart Lung Innovation, St. Paul’s Hospital, Vancouver, BC V6Z 1Y6, Canada

**Keywords:** CFTR modulator, ivacaftor, lumacaftor, tezacaftor, elexacaftor, real-world, adverse events, safety

## Abstract

Cystic fibrosis transmembrane conductance regulator (CFTR) modulator therapies target the underlying cause of cystic fibrosis (CF), and are generally well-tolerated; however, real-world studies indicate the frequency of discontinuation and adverse events (AEs) may be higher than what was observed in clinical trials. The objectives of this systematic review were to summarize real-world AEs reported for market-available CFTR modulators (i.e., ivacaftor (IVA), lumacaftor/ivacaftor (LUM/IVA), tezacaftor/ivacaftor (TEZ/IVA), and elexacaftor/tezacaftor/ivacaftor (ELX/TEZ/IVA)), and to identify ways in which the pharmacist on CF healthcare teams may contribute to mitigating and managing these AEs. The MEDLINE, EMBASE, CINAHL, and Web of Science Core Collection online databases were searched from 2012 to 1 Aug 2020. Full manuscripts or conference abstracts of observational studies, case series, and case reports were eligible for inclusion. The included full manuscripts and conference abstracts comprised of 54 observational studies, 5 case series, and 9 case reports. The types of AEs reported generally aligned with what have been observed in clinical trials. LUM/IVA was associated with a higher frequency of respiratory-related AE and discontinuation in real-world studies. A signal for mental health and neurocognitive AEs was identified with all 4 CFTR modulators. A systematic approach to monitoring for AEs in people with CF on CFTR modulators in the real-world setting is necessary to help better understand potential AEs, as well as patient characteristics that may be associated with higher risk of certain AEs. Pharmacists play a key role in the safe initiation and monitoring of people with CF on CFTR modulator therapies.

## 1. Introduction

Cystic fibrosis (CF) is an autosomal recessive disorder affecting the *CF transmembrane conductance regulator (CFTR)* gene, resulting in alteration of CFTR protein synthesis, processing, or function. The CFTR protein is a channel that transports chloride and bicarbonate across cell membranes, regulating fluid and electrolyte balance in several organs throughout the body; dysfunctional CFTR protein results in viscous mucus and secretions [1]. In the lungs, this viscous mucus causes impaired mucociliary clearance as well as chronic infection and inflammation, which, over time, leads to irreversible airway damage and progressive lung function decline [1]. The pancreas, gastrointestinal tract, and biliary ducts are among the extrapulmonary organs impacted by CF, resulting in complications such as nutrient malabsorption, bowel obstruction and hepatobiliary disease, respectively [1]. Optimal management of CF requires a complex, multimodal medication regimen that may take upwards of 2 h to administer on a daily basis [2]. Pharmacists have therefore been identified as integral members of multidisciplinary teams providing care to people with CF, with essential roles such as: optimizing the effectiveness of medication regimens and tailoring to each individuals’ needs; providing medication education and counselling; promoting adherence to therapies; as well as monitoring and managing drug-related adverse events (AE) and interactions [3,4,5,6].

Prior to the development of CFTR modulator therapies, available treatments for CF only targeted symptom management, whereas CFTR modulators target the underlying cause [1,7]. CFTR modulators are small molecule therapies, with four single or combination therapies currently available on the market: ivacaftor (IVA; Kalydeco^®^), lumacaftor/ivacaftor (LUM/IVA; Orkambi^®^), tezacaftor/ivacaftor (TEZ/IVA; Symdeko^®^ or Symkevi^®^), and elexacaftor/tezacaftor/ivacaftor (ELX/TEZ/IVA; Trikafta^®^ or Kaftrio^®^) [8,9,10,11,12,13]. IVA was the first CFTR modulator brought to market, approved by the Food and Drug Administration (FDA) in 2012; it is a CFTR “potentiator” that increases the amount of time the CFTR protein channel remains open, targeting *CFTR* mutations that impact channel gating and conductance (e.g., *G551D, R117H)* [8]. The remaining three CFTR modulator therapies combine IVA with one or two CFTR “correctors”. “Correctors” primarily target the most common *CFTR* mutation, *F508del*, improving CFTR protein conformation and subsequent processing and trafficking to the cell surface [9,10,11,12,13]. Results from randomized, placebo-controlled clinical trials support the efficacy of CFTR modulator therapies in the treatment of people with CF: IVA in patients with at least one *G551D* or *R117H* mutation*,* non-*G551D* gating mutation, or residual function mutation [14,15,16,17,18]; LUM/IVA in patients who are *F508del* homozygous [19,20]; TEZ/IVA in patients who are *F508del* homozygous or heterozygous with a residual function mutation [18,21]; and ELX/TEZ/IVA in patients who are *F508del* homozygous or heterozygous with a minimal function mutation [22,23].

In the aforementioned clinical trials, CFTR modulators were generally well-tolerated, with the exception of LUM/IVA having higher rates of respiratory-related AE [7,19,20]. However, observational studies with real-world CFTR modulator safety data have raised flags for higher rates of discontinuation as well as AE that were rarely observed or not described in the clinical trial setting. The objective of this systematic review was to summarize real-world AEs reported for CFTR modulator therapies, as well as to identify ways in which the pharmacist on CF healthcare teams may contribute to mitigating and managing these AEs.

## 2. Methods

### 2.1. Search Strategy

The MEDLINE, EMBASE, CINAHL, and Web of Science Core Collection online databases were systematically searched from 2012 to 1 August 2020 to identify literature related to the primary research question. The search terms and Boolean operators utilized in MEDLINE and EMBASE are outlined in Appendix A as an example. References of relevant literature were searched manually to identify additional studies not identified in the electronic search, and grey literature was explored via conference abstracts identified in the search. The systematic review protocol was prospectively registered with PROSPERO (CRD42020188189).

### 2.2. Selection of Studies

Observational studies, case series, and case reports were eligible for inclusion in the review if study participants had a diagnosis of CF, received at least one dose of a market-available CFTR modulator (i.e., IVA (Kalydeco^®^), LUM/IVA (Orkambi^®^), TEZ/IVA (Symdeko^®^ or Symkevi^®^), or ELX/TEZ/IVA (Trikafta^®^ or Kaftrio^®^) [8,9,10,11,12,13]) in the real-world setting, and reported AEs that occurred while participants were receiving the CFTR modulator. Conference abstracts that met the above criteria were also included. Studies were excluded if it was a clinical trial or study utilizing clinical trial data, was not available in the English language, if none of the AE were specified, or if it was reported that no AEs occurred in the study participants. Two authors (R.D. and V.S.) independently conducted a search and reviewed potentially relevant citations to determine whether the pre-defined criteria for inclusion were satisfied; any inconsistencies in studies selected for inclusion were resolved by discussion and consensus.

### 2.3. Data Extraction and Management

Data from each included study were extracted and tabulated. Data extracted from full manuscripts included: first author; year of publication; study design; study location; study population age group, *CFTR* genotype, and baseline percent predicted of the forced expiratory volume in the first second (ppFEV_1_); CFTR modulator(s) received by study participants; follow-up time of study participants; type and frequency of AEs; CFTR modulator interruption, discontinuation, and/or dose modification secondary to AEs; and strategies to address or mitigate reported AEs (if described). Data extracted from conference abstracts was the same as for full manuscripts, with the exception of study location and follow-up time, as these were infrequently specified. Data were first extracted by the primary reviewer (R.D.) and checked by the secondary reviewer (V.S.); disagreements or inconsistencies regarding extracted data were resolved by discussion and consensus.

### 2.4. Quality Assessment

All full manuscripts included in the review were independently assessed for quality by the primary and secondary reviewer (R.D. and V.S.) using a National Institutes of Health, National Heart, Lung, and Blood Institute Quality Assessment Tool appropriate for the study design [24]. The quality of the manuscripts was assessed from the perspective of addressing safety events (i.e., a study may have been methodologically sound for assessment of effectiveness or other outcomes unrelated to safety but may have received a lower rating for this systematic review due to seemingly not having a robust approach to assessing safety outcomes). An overall rating of ‘Good’, ‘Fair’, or ‘Poor’ was assigned to each manuscript after discussion and consensus based on the assessment tool criteria and study-specific factors that may affect the quality and internal validity. Case reports were not systematically assessed for quality due to absence of a standardized quality assessment tool, and conference abstracts were not formally assessed due to inherent lack of detail necessary to appropriately assess the quality of study methodology.

### 2.5. Data Analysis

Descriptive analyses were performed on data extracted from the included studies.

## 3. Results

### 3.1. Description of Studies

Of the 2333 articles identified in the search or through other sources, 278 were screened for eligibility. Of the 278 screened, 31 studies with full manuscripts and 37 conference abstracts met the inclusion criteria, and 210 articles were excluded (Figure 1). Of the 31 full manuscripts, 20 were observational studies, 5 were case series, and 6 were case reports. Of the 37 conference abstracts, 34 were observational studies and 3 were case reports. Characteristics of the full manuscripts and conference abstracts describing observational cohort studies are summarized in Table 1 and Table 2, respectively. Two observational cohort studies included control subjects as comparison for non-safety outcomes related to LUM/IVA therapy [25,26], but only data regarding the participants on LUM/IVA was extracted and included in the descriptive analysis of this review. One case series was based on survey results presented in aggregate [27]; therefore, it was included in Table 1. Characteristics of case series and case reports are otherwise summarized in Table 3.

The majority of the included studies and case series/reports regarded LUM/IVA, totaling 47 (69%) of the 68 included. IVA, TEZ/IVA, and ELX/TEZ/IVA were the subject of 11 (16%), 7 (10%), and 4 (6%) of the included manuscripts and abstracts, respectively (one study discussed both LUM/IVA and TEZ/IVA, so was included in the totals for both modulators [28]). Eleven (16%) of the included studies and case series/reports included only pediatrics (age < 18 years old), 33 (49%) included only adults, 16 (24%) included both adult and pediatric patients, and the age groups included was not specified in 8 (12%). In the observational studies, the number of subjects included ranged from 4 to 845 (median: 28). Fourteen (26%) of the observational studies focused on patients with severe airflow obstruction, defined as ppFEV_1_ < 40%; in 11 of these studies, patients accessed the CFTR modulator via a compassionate access program (also referred to as a “managed access”, “early access’, “expanded access”, or “named patient” program) through the manufacturer [25,26,29,30,31,32,33,34,35,36,37,38,39,40]. Twenty (65%) of the full manuscripts were deemed to have safety as a primary focus or outcome [27,28,29,30,32,33,41,42,43,44,45,46,47,48,49,50,51,52,53,54].

### 3.2. Quality Assessment

The 20 observational studies and 5 case series with full manuscripts were assessed for quality from the standpoint of AE follow-up and assessment. Ten of the observational studies were rated as ‘Good’ quality, eight were rated as ‘Fair’, and two were rated as ‘Poor’ (Appendix B). Only one of the case series was rated as ‘Good’ quality, while one was rated as ‘Fair’ and three were rated as ‘Poor’ (Appendix C). The two leading factors contributing to lower quality ratings were unclear and/or inconsistent methods for follow-up and assessment of AEs, and lack of a reliable, validated tool or method to accurately assess and confirm a given AE.

### 3.3. AE-Related Outcomes

The AE reported for IVA, LUM/IVA, TEZ/IVA, and ELX/TEZ/IVA are summarized in Table 1, Table 2 and Table 3.

#### 3.3.1. Ivacaftor (IVA)

In the studies evaluating IVA, a large proportion of the reported AEs were respiratory-related. In two observational studies that reported serious respiratory-related AEs such as pulmonary exacerbation or respiratory infection, hemoptysis, and acute respiratory failure, these respiratory AEs did not prompt discontinuation and were attributable to patients’ underlying disease [29,55]. Another study reported increased bronchial and nasal secretions after initiating IVA in a patient with low lung function (baseline ppFEV_1_ 13.9%) prompting a severe pulmonary exacerbation, and IVA ultimately had to be discontinued due to difficulty clearing “liquefied” secretions despite decreases in dose [30]. Otherwise, respiratory AE reportedly resolved over time without changes to IVA therapy [30,46].

Relatively common were transaminitis and other hepatic AEs [29,30,55,56,58,64]. In one observational study, transaminitis and a diagnosis of liver cirrhosis resulted in IVA discontinuation in one individual each, and hepatitis warranted interruption of therapy [58]. One case report indicated a decrease in IVA dose was necessary for normalization of transaminases [46]. Hepatic AEs were not otherwise reported to require changes in IVA therapy.

Abdominal pain was reported in a number of studies, though the overall frequency was often not specified [29,30,58]. In some cases, abdominal pain was severe enough to warrant interruption or discontinuation of therapy [29,58]. Nausea or vomiting, intestinal dysmotility, and gastroenteritis were other gastrointestinal-related AEs reported, of which vomiting was the only one to prompt interruption of therapy [29,58].

Headache and rash were also common, but only prompted IVA interruption in one study [29,30,55,57,58,62,63,65]. Less common but notable AEs included cataracts, depression, dizziness, and tinnitus; the latter two prompted discontinuation of IVA in one patient, while depression warranted interruption of therapy [29,55,57,58,65]. IVA was also implicated in causing recurrent morbilliform drug eruption (Type IV drug hypersensitivity) to trials of LUM/IVA and TEZ/IVA, with successful IVA desensitization thereafter [28].

#### 3.3.2. Lumacaftor/Ivacaftor (LUM/IVA)

The majority of AEs reported with LUM/IVA were respiratory-related. Chest tightness, dyspnea, increased sputum, and declines in ppFEV_1_ were among the most common respiratory AEs and tended to occur within the first few days after initiation, even as quickly as 3–4 h after the first dose [25,26,27,31,32,33,34,35,37,38,41,42,43,44,45,60,61,66,67,68,69,70,71,72,73,74,75,76,77,78,83,84,85,90]. Bronchodilators were beneficial in mitigating symptoms of chest tightness, wheeze, and increased work of breathing in some individuals [31,41,66,67,75]. Improvement in or resolution of respiratory AEs generally occurred over the 1–4 weeks following initiation, but symptoms and/or ppFEV_1_ below baseline could persist beyond this time [26,31,32,34,42,43,73]. In studies assessing ppFEV_1_ changes post-initial dose of LUM/IVA, a mean absolute decline in ppFEV_1_ around 10% within 4 h was common [41,43,67], with declines of up to 20% at 4 and 24 h reported even when bronchodilators were administered pre-dose [43]. One study reported a ppFEV_1_ drop 39% below baseline in an individual; the timing of repeat spirometry after LUM/IVA initiation was not specified [44]. Declines in ppFEV_1_ were not always accompanied by respiratory symptoms, and bronchodilators did not consistently improve acute drops in ppFEV_1_ [32,41,43,66,67,84]. ppFEV_1_ could take upwards of three months after initiation to recover to baseline values [34,43]. Increased cough was common, sometimes resulting in discontinuation [33,42,44,45,69,73,76,85], while chest pain was infrequently reported [37,45,72,85]. Two cases of hemoptysis and one case of pneumothorax reportedly caused discontinuation of LUM/IVA in one study; whether the hemoptysis and pneumothorax were attributable to underlying disease was not specified [42].

Intolerable respiratory AE could be overcome in some patients by reducing the dose of LUM/IVA [66,70]; however, discontinuation due to respiratory AEs remained common. Chest tightness and/or dyspnea were the most common respiratory AEs to prompt discontinuation, with reported frequencies between from 5 to 31%. In one case report, halving the LUM/IVA dose mitigated severe chest tightness, dyspnea, and exertional hypoxemia that onset after initiation, but therapy was ultimately discontinued with rapid recovery of symptoms and ppFEV_1_ thereafter [90]. A cohort study also reported resolution of respiratory AE upon LUM/IVA discontinuation [35]. In observational studies that described retrials after LUM/IVA was discontinued, there were mixed outcomes among patients. In one study, 32 of the 90 patients who discontinued LUM/IVA at least once due to respiratory AE were retrialed—16 were able to successfully restart therapy, and 16 had to discontinue indefinitely [42]. Three studies also described LUM/IVA retrial following discontinuation due AE: in the first, of 25 patients who discontinued, 9 were retrialed, and 6 had recurrence of the same AE [72]; in the second, 4 of 8 patients who discontinued were retrialed at full- or half-dose and experienced AE recurrence [76]; and in the third, 6 of 8 patients who discontinued were retrialed and able to restart therapy, but 4 required half-dose in order to do so [78]. The three studies were reported in abstracts and did not specify whether it was only patients with respiratory AE that were retrialed, but respiratory AE accounted for a large proportion of the AEs prompting discontinuation.

A number of factors and patient characteristics were explored as potential risk factors for respiratory AE and/or discontinuation due to AE with LUM/IVA. Having a lower baseline ppFEV_1_ was generally associated with greater risk of respiratory AE or discontinuation [26,42,73,75,77], but differences were not always statistically significant [33,35,44,61], and one study found the contrary [43]. One study, using multivariable logistic regression, found for every 10% decrease in baseline ppFEV_1_, there was a greater likelihood of LUM/IVA discontinuation overall and due to respiratory AE—odds ratio (OR) 1.13 (95%CI, 1.02–1.25; *p* = 0.02) and OR 1.32 (95%CI, 1.14–1.51; *p* = 0.0001), respectively [42]. Older age and greater use of IV antibiotics in the past year had conflicting results for whether they were significant risk factors for LUM/IVA discontinuation [42,44,61]. The association with reactive airway disease and/or atopy was discussed in three studies. Reversible airway obstruction ≥12%, but not personal or familial history of atopy, was significantly associated with a greater drop in ppFEV_1_ when evaluated in one study [41], and higher frequency of discontinuation or drop in ppFEV_1_ was found in patients with asthma or asthma/atopy, respectively [67,77]; whether statistical significance was reached in the latter two studies was not specified. Interestingly, higher rates of discontinuation overall were associated with female sex in one study (adjusted OR 3.12 (95%CI, 1.04–9.38; *p* = 0.04) [44]), but this was not observed in another evaluating this relationship [61]. Discontinuation due to respiratory AE was associated with diabetes (OR 1.71 (95%CI, 1.03–2.85; *p* = 0.04)) and low body mass index (BMI) (OR 1.11 (95%CI, 1.00–1.23; *p* = 0.03) per 1 kg/m^2^ decrease) [42].

Similar to IVA, hepatic AE were relatively common with LUM/IVA. Transaminitis and other liver function test (LFT) elevations were seemingly mild and/or transient without need for intervention for the most part, but sometimes warranted discontinuation [34,42,45,59,60,61,72,76,78,81]. One study specified that transaminases were over 6 times the upper limit of normal when discontinued [42]. A case of isolated alkaline phosphatase elevation was also reported, taking several months for the alkaline phosphatase to normalize after discontinuation of LUM/IVA [91].

Similarly common with LUM/IVA were gastrointestinal AEs, of which abdominal pain, nausea and/or vomiting, and diarrhea were the most frequently reported and resulted in discontinuation in two studies each [26,33,34,42,44,45,61,69,71,73,78,79,85]. Uncommon gastrointestinal AEs were dysphagia and decreased appetite; the one case of dysphagia resulted in discontinuation [44,61].

Rash and hypersensitivity reactions were notable with LUM/IVA. In several observational studies reporting rash, LUM/IVA was seemingly continued without intervention, but prompted discontinuation in a small number of patients [26,33,35,42,44,73,76,78]. More serious reactions in the observational studies were rash with facial swelling, allergic reaction with chest tightness, and suspected Stevens–Johnson syndrome (SJS), reported in one individual each [61,75]. The LUM component of LUM/IVA was implicated via in vitro testing to have triggered a T-cell-mediated reaction that caused a progressive severe, pruritic rash and facial swelling in one case report [50]. Another case report described morbilliform drug eruption recurring on three separate trials of LUM/IVA, but IVA was deemed the likely culprit in the reactions [28]. In both case reports, the hypersensitivity reactions resolved quickly following discontinuation [28,50].

An important signal was noted for the impact of LUM/IVA on mental health. One large prospective cohort study reported four individuals (0.5% of the study cohort) discontinued LUM/IVA due to depression [42], and one individual in each of two smaller retrospective cohorts reported anxiety as the cause of discontinuation (4 and 5% of the respective cohorts) [45,75]. One case series described five adolescent females, comprising 24% of the adolescent females who initiated LUM/IVA at the reporting center, with new or worsening depression as early as two weeks following LUM/IVA initiation [47]. Two of the patients had suicidal ideation (one each with baseline and new-onset depression), while three of the patients had an attempted suicide requiring hospitalization (one with baseline and two with new-onset depression) [47]. In all but one patient, mood improved after discontinuing LUM/IVA [47]. One of the patients with new-onset depression and suicidal ideation also had worsening of baseline anxiety, which, along with menstrual irregularity, was the ultimate cause of LUM/IVA discontinuation [47]. A second case series described four individuals with worsening of their baseline psychiatric diagnoses, including depression, anxiety, bipolar disorder, and/or substance use disorder, as early as one month following LUM/IVA initiation [48]. In three individuals, worsening mental health triggered a clinical decline and pulmonary exacerbation, likely attributable to medication non-adherence [48]. One patient discontinued LUM/IVA due to worsening depression and anxiety, and required adjustment of psychotropic therapy to return to baseline [48]. The other two patients remained on LUM/IVA; for one patient, the mood returned to baseline only after adjustment of psychotropic medications and the other patient did not return to baseline levels of anxiety despite appropriate treatment interventions [48]. The final patient discontinued LUM/IVA due to non-adherence [48].

Two uncommon but consequential AEs were elevations in blood pressure and creatine kinase (CK). In 1 cohort of 22 patients, 5 (23%) had elevations in blood pressure, of whom 4 discontinued [45]. Two of the patients had symptomatic hypertension: one presented with headache and required two antihypertensives to manage her blood pressure before LUM/IVA was stopped, and the second had a hypertensive emergency that onset within 12 h of his first LUM/IVA dose (notably, he was initiated at one-quarter dose due to a drug–drug interaction with posaconazole) [45]. One of the patients with asymptomatic hypertension remained above his baseline blood pressure even after LUM/IVA discontinuation [45]. In a cohort of 30 patients admitted to hospital for LUM/IVA initiation, 1 (3%) discontinued due to hypertension; the onset of the hypertension was seemingly quick, as it occurred before the patient was discharged from hospital [37]. An elevation in CK was reported in a large cohort study wherein 20 (2%) patients had a CK greater than five times the upper limit of normal, and five (0.6%) had to discontinue LUM/IVA due to a CK greater than 10 times the upper limit of normal accompanied by myalgia [42]. One case report also describes an adult male who experienced rhabdomyolysis with a CK upwards of 17,000 U/L, attributed to a drug–drug interaction between LUM/IVA and acyclovir; despite normalization of the CK following discontinuation of both LUM/IVA and acyclovir, lower limb myalgia persisted [92].

Headache and fatigue were reported relatively often overall but were not a common cause of LUM/IVA discontinuation [25,26,33,34,42,45,69,73,76]. Menstrual irregularities and menorrhagia were reported infrequently but did result in discontinuation of LUM/IVA—the former in one patient who also discontinued due to worsening anxiety [47], and the latter in three individuals (0.4%) in a large cohort [42]. Uncommon AEs of note were cataracts, which prompted discontinuation in one individual [60]; one case each of bradycardia and tachycardia, the latter of which prompted discontinuation [42,45]; and a case of acute myelocytic leukemia (AML) post-partum [27]. The authors of the case series reporting AML deemed it likely an unfortunate coincidence, unrelated to LUM/IVA [27].

#### 3.3.3. Tezacaftor/Ivacaftor (TEZ/IVA)

Limited data were available regarding real-world AEs with TEZ/IVA. Notable were neuropsychiatric AEs. In a cohort of 44 adults, 5 (11%) experienced neuropsychiatric AEs: out of body experience and visual hallucination; depersonalization and “brain fog”; severe migraine; and sleep pattern disturbance, which occurred in 2 individuals [88]. The two individuals with sleep pattern disturbance remained on TEZ/IVA, but the remaining three individuals changed back to LUM/IVA and had resolution of symptoms thereafter [88]. Another cohort study reported discontinuation due to unspecified mental health changes; the frequency was not reported [86].

A case study described morbilliform drug eruption that recurred on retrial of TEZ/IVA and resolved upon discontinuation; however, IVA was deemed the likely cause of the reaction, and after a stepwise desensitization to IVA the patient was able to resume TEZ/IVA [28]. One other cohort reported rash in two (9%) individuals that did not require discontinuation [39]. Liver enzyme abnormalities or elevations resulted in TEZ/IVA discontinuation in two cohort studies [86,87], while hair loss and fatigue, and blurred vision were the cause of TEZ/IVA discontinuation in one individual each [39,89]. Acholic stools, persistent nausea and vomiting, changes in blood glucose, and new-onset hemoptysis were also reported to require discontinuation of TEZ/IVA, but the frequency was not specified [86].

One case report describes a delayed suspected drug–drug interaction between TEZ/IVA and azithromycin [51]. In an adult female with no history of cardiac arrhythmia and who had been on long-term therapy with azithromycin prior to TEZ/IVA initiation, an electrocardiogram (ECG) at 36 weeks of concomitant therapy revealed first-degree heart block (PR interval of 334 ms); ECGs at baseline and 4, 12, and 24 weeks after initiating TEZ/IVA were normal [51]. The first-degree heart block resolved by four weeks after discontinuation of TEZ/IVA, and therapy was retrialed eight months later. Baseline and follow-up ECGs at 4, 8, 12, and 24 weeks were normal, but an ECG at 39 weeks revealed recurrence of the first-degree heart block (PR interval of 215 ms) [51]. Azithromycin was discontinued and the PR interval normalized by four weeks thereafter [51].

#### 3.3.4. Elexacaftor/Tezacaftor/Ivacaftor (ELX/TEZ/IVA)

Given the relatively recent market approval in some countries, limited data were available regarding real-world AEs with ELX/TEZ/IVA. One case series reported on seven adults who experienced biliary colic following initiation of ELX/TEZ/IVA; all but one patient received a cholecystectomy [53]. Only one of the seven patients was CFTR modulator naïve prior to ELX/TEZ/IVA, and five of the patients had chronic cholecystitis and cholelithiasis [53]. In those receiving a cholecystectomy, ELX/TEZ/IVA remained unchanged in four patients and was briefly held perioperatively in two patients due to severe symptoms or postoperative complications; symptoms resolved in all six [53]. ELX/TEZ/IVA was held two weeks before and four weeks after the cholecystectomy in the fourth patient, and based on personal communication with the corresponding author, although symptoms of biliary colic resolved, ELX/TEZ/IVA was subsequently held due to transaminitis [53].

A second case series described testicular pain that onset shortly after ELX/TEZ/IVA initiation in seven males [52]. For five patients, ELX/TEZ/IVA continued uninterrupted—three had no interventions and two required over-the-counter analgesics to manage pain [52]. One patient required a brief interruption in therapy and restarted at full-dose, while the final patient required a dose reduction and titration back to full-dose in addition to over-the-counter analgesics and antibiotics for epididymoorchitis and scrotal wall cellulitis [52]. All patients had resolution of the testicular symptoms, and at least three patients also experienced abdominal bloating or constipation [52]. Transaminitis was reported in four (36%) patients in a cohort study, but intervention was not required [40].

ELX/TEZ/IVA was also associated with mental health-related AE, as described in a case report of a 19-year-old female [54]. In the described case, the patient experienced worsening of baseline depression and sleep paralysis with hypnopompic hallucinations, and new passive suicidal ideation following initiation of ELX/TEZ/IVA [54]. The latter resolved upon discontinuation, but initiation of an antidepressant was necessary to improve mood [54]. Over the course of several months, the ELX/TEZ/IVA dose was titrated, and dosage times changed in response to worsening mood, suicidality, anxiety, and sleep paralysis. Although interpersonal conflicts may have affected symptoms, the reported symptoms were seemingly worst at the full-dose of ELX/TEZ/IVA and improved with dose reductions [54]. The dose eventually settled on two ELX/TEZ/IVA tabs once daily in the morning, with depression and anxiety symptoms above baseline and recurrent episodes of sleep paralysis with hypnopompic hallucinations [54].

#### 3.3.5. Described Strategies to Address or Mitigate Reported Adverse Events

Protocols and strategies utilized in studies and/or suggestions that arose as a result of observed study outcomes are summarized in Table 4. To minimize respiratory-related AE and risk of discontinuation with LUM/IVA, two studies implemented an initiation protocol [31,43], while another study informally made efforts to optimize inhaled therapies in patients with ppFEV_1_ ≤ 40% [44]. A fourth study established a desensitization protocol for IVA and transition to TEZ/IVA following recurrent morbilliform drug eruption [28]. The majority of suggestions that resulted from study observations pertained to LUM/IVA, primarily in relation to respiratory AE [25,26,32,33,41,49,61,75], as well as new or worsening depression and anxiety [47,48], hypertension [45], and the potential drug–drug interaction with PPIs [79]. Three studies provided suggestions for ELX/TEZ/IVA based on direct observations—one each for testicular pain, biliary colic, and worsening mental health—and a fourth based indirectly on observed delayed-onset first-degree heart block with concomitant TEZ/IVA and azithromycin [51,52,53,54]. In addition to the latter study, suggestions for TEZ/IVA were provided based on observed neurocognitive AEs [88]. Suggestions for IVA regarded initiation in patients with severe lung disease and monitoring of pediatric patients for cataract formation [30,57].

## 4. Discussion

### 4.1. Putting Real-World Adverse Events into Context

The primary objective of this systematic review was to evaluate the real-world AE profile of currently-marketed CFTR modulator therapies in the treatment of people with CF, with a secondary objective of evaluating reported strategies to address or mitigate reported AEs.

When interpreting the real-world AE findings in the context of clinical trial data, it is prudent to keep in mind that study populations in clinical trials are fundamentally different from real-world populations due to reasons such as strict inclusion/exclusion criteria and participants being inherently more motivated to continue with the assigned therapy [93]. In clinical trials evaluating CFTR modulators, study participants were clinically stable and those with severe or minimal lung disease (i.e., ppFEV_1_ < 40% and > 90%, respectively) were typically excluded or grossly underrepresented [94,95]. Another important consideration is the differences between observational studies and clinical trials in AE evaluation and reporting. Clinical trials have more rigorous monitoring for AEs, increasing the likelihood of identifying AEs, and AEs are reported even if not attributed to the intervention; this is demonstrated in 3 clinical trials evaluating CFTR modulators, wherein an absolute difference in the frequency of AE overall and AE deemed related to trial drug ranged from 12 to 53% within the treatment arms [18,22,96]. That said, patients and study investigators may also be less inclined to report AEs or attribute an AE to study drug to avoid the patient being removed from the trial, especially if the patient has experienced a dramatic clinical benefit. Any relative comparisons made hereinafter are general and should be interpreted with the aforementioned considerations in mind.

For IVA and LUM/IVA, the majority of reported AE were reflective of what may be expected based on the findings in clinical trials; the same comparison could not be made for TEZ/IVA and ELX/TEZ/IVA due to the limited published real-world data available. Of the four CFTR modulators, LUM/IVA was seemingly associated with a higher frequency of AE, respiratory-related AE in particular. While this observation may be due to real-world studies predominantly reporting on experience with LUM/IVA, it reflects what has also been observed in the clinical trial setting [7,94,95]. Dyspnea and chest tightness with LUM/IVA appear to have occurred more often in real-world studies, with higher frequencies overall and of discontinuation than reported in clinical trials. This is likely due to the larger proportion of patients with severe lung disease in real-world studies. In fact, when Burgel et al. re-examined the data from their prospective cohort study [42], they found that in patients who discontinued LUM/IVA, respiratory AE were the cause in 74% of patients with a baseline ppFEV_1_ < 40%, compared to 42% and 29% in patients with a baseline ppFEV_1_ of 40–90% and ≥ 90%, respectively [97]. However, even patients with a baseline ppFEV_1_ ≥ 40% had overall LUM/IVA discontinuation rates upwards of threefold higher than in the clinical trial setting [42]. Moreover, in the TRAFFIC/TRANSPORT trials, while the frequency of dyspnea was higher in the pooled LUM/IVA group compared to placebo overall, the frequency in study participants with a baseline ppFEV_1_ < 40% was approximately twice that of those with a baseline ppFEV_1_ ≥ 40% [98]. It should be noted that although there is greater reporting of respiratory-related AEs in patients with more advanced lung disease on LUM/IVA, asymptomatic acute drops in ppFEV_1_ were also common in pediatric patients with milder lung disease [41,43].

The mechanism behind the respiratory-related AE reported for LUM/IVA has not been fully elucidated but is unlikely explained by CFTR modulation alone as it has not been described with other more effective CFTR modulators. Furthermore, this effect is likely specific to LUM as it was not described with IVA monotherapy. Bronchoconstriction is a likely contributor, with the mitigation of respiratory AE observed with bronchodilators in real-world studies; this was also demonstrated in an open-label study in healthy subjects, wherein administration of a short-acting bronchodilator reversed the drop in ppFEV_1_ observed at 4 h post-LUM/IVA dose, and long-acting bronchodilator administered 12 h pre-dose diminished the drop in ppFEV_1_ [99]. The incidence of respiratory-related AE with LUM/IVA is seemingly highest around initiation of therapy, justifying suggestions to initiate at lower doses, especially in those with lower lung function. Despite efforts to optimize inhaled therapy and initiate LUM/IVA at lower doses for at least a week in patients with ppFEV_1_ ≤ 40%, Jennings et al. [44] still observed higher frequencies of respiratory-related AE in this subgroup; however, it is possible the frequency would have been higher without these precautions, which was the case in an open-label study of LUM/IVA by Taylor-Cousar et al. [100]. In this study involving patients with advanced CF lung disease, treating physicians were permitted to initiate patients at half-dose for up to 15 days and titrate thereafter to full-dose; those initiated at half-dose had a lower incidence and duration of respiratory-related AE, and unlike those initiated at full-dose, respiratory-related AE did not necessitate discontinuation or dose modification of LUM/IVA in those initiated at half-dose [100]. Moreover, Murer et al. [31] attributed the relatively low frequency of discontinuation due to respiratory-related AE in their single-center study of patients with severe CF-related lung disease to their step-wise LUM/IVA initiation protocol. Burgel et al. [42] reported higher discontinuation rates in those who initiated LUM/IVA at a reduced dose instead of full dose (25% vs. 17.3%, respectively), but this difference was not statistically significant and the majority of patients initiating at lower doses were due to potential drug interactions, not necessarily precautions due to low lung function. Further, in the case that lower initiation doses were utilized in patients deemed higher risk for respiratory-related AE, the higher discontinuation rate could be a reflection of confounding by indication.

When considering the factors found to be associated with increased risk for AE or discontinuation of LUM/IVA, increased risk of respiratory AE and discontinuation due to AE in patients with lower lung function is expected from what has been observed in the literature. Accordingly, so is more frequent need for IV antibiotics, which may be a reflection of more advanced lung disease, as well as CF-related diabetes and low BMI, as both are correlated and associated with worse pulmonary function [101,102]. Older age as a risk factor may also be a reflection of lower lung function by nature of CF being a progressive disease over time, but the basis of female sex as a risk factor is unclear, and the authors too could not rationalize why this was observed [44]. With bronchoconstriction as a likely contributing mechanism in LUM/IVA respiratory-related AE, reversible airway obstruction ≥12% as a risk factor is also justifiable, and although the studies evaluating asthma with or without atopy did not report on statistical significance, the sample sizes were likely too small to detect one [67,77].

Rash was not uncommon, reported in real-world studies for each of IVA, LUM/IVA, and TEZ/IVA, with few individuals requiring interruption or discontinuation of therapy for rash or allergic reactions. Similar was seen in clinical trials, with cases of rash being reported for all four CFTR modulators, and serious rash or discontinuation due to rash being reported for ELX/TEZ/IVA [22,23,103] and LUM/IVA [19,104]. Anecdotally at our site, one case of suspected anaphylaxis occurred following the first dose of TEZ/IVA; no CFTR modulators have ever been retrialed in this patient. Given no additional information regarding the suspected case of SJS was provided [61], it is not possible to evaluate whether it was potentially attributable to LUM/IVA; however, with the report of the severe delayed CD4+ lymphocyte-mediated reaction to LUM/IVA [50], the possibility for SJS and other severe Type-IV (i.e., delayed) hypersensitivity reactions to CFTR modulators is not inconceivable. Unlike the described morbilliform drug reaction with successful desensitization to IVA [28], it would be inappropriate to attempt desensitization to these more serious delayed hypersensitivity reactions [105]. In a phase 3 clinical trial of ELX/TEZ/IVA, the incidence of rash was highest in females, particularly those on hormonal contraceptives [23]; the basis of the increased incidence of rash in either of these subgroups is unclear.

An important signal of AE identified in the real-world studies that were uncommon or not reported in the clinical trials was related to mental health and neurocognitive or neuropsychiatric events. Though evidence-based interventions were also implemented in several cases, discontinuation or dose adjustment of the CFTR modulator seemingly resulted in symptom improvement for some individuals [47,48,54]. In PERSIST, the open-label extension study of the STRIVE and ENVISION trials, one study participant discontinued IVA due to depression and one suicide occurred; the latter was deemed unlikely related to IVA [106]. Serious AE reported in one individual each were affective disorder, depression, major depression, suicidal ideation, and suicidal depression, but whether or not these were attributable to IVA was not stated [106]. Though not published in the original manuscripts, McKinzie et al. [47] reported that a number of cases of depression, depressed mood, or anxiety were reported in the TRAFFIC/TRANSPORT trials and PROGRESS extension study for LUM/IVA; for two participants in the TRAFFIC/TRANSPORT trials the symptoms were reportedly considered related to study drug, but this was not specified for the PROGRESS study. Though causation cannot be attributed to CFTR modulators it is prudent to consider the potential for these medications to affect mental health outcomes, as depression has been associated with accelerated lung function decline in adolescents and adults with CF [107] as well as increased five-year mortality in adults with CF, particularly in those with severe depression [108]. Indeed, in three of the reported cases there was significant lung function decline following worsening mental health with LUM/IVA, potentially due to the negative impact of mood on adherence to maintenance therapies [48]. The mechanism by which CFTR modulators may impact mental health is not well-understood; induction of cytochrome P450 (CYP) enzymes by LUM and resultant decreased exposure to substrate psychotropic medications is one potential contributor [9,109], but this does not account for the other CFTR modulators which do not share this drug–drug interaction or individuals who do not have a known history of mental health concerns and/or who are not on a psychotropic medication.

A potential mechanism by which ELX/TEV/IVA (or other CFTR modulators) may trigger biliary colic was hypothesized by Safirstein et al. [53]; restoring CFTR function in the biliary epithelium will cause changes in the fluidity and acidity of bile fluids, which may result in mobilization of existing gallstones and precipitation of biliary colic. This hypothesis has merit, given the underlying pathophysiology of CF-related hepatobiliary disease [110]. That restoration of CFTR function may also dislodge mucus blockages from the testes and/or vas deferens was posed by Rotolo et al. [52] as a potential the mechanism behind testicular pain in males following initiation of ELX/TEZ/IVA. One serious AE that was not mentioned in the included real-world studies but warrants discussion is distal intestinal obstruction syndrome (DIOS). DIOS has been reported in clinical trials for all four CFTR modulators, albeit at a low frequency [15,18,19,96,103,106,111,112,113,114]. Although DIOS may be secondary to the underlying CF itself, there is a plausible mechanism for how DIOS may be of concern with the CFTR modulators, particularly at the time of initiation. Chronic constipation is not uncommon in CF, with viscous intestinal mucus and gastrointestinal dysmotility resulting in undigested food adhering to the intestinal walls [115]. Theoretically then, with initiation of a highly effective CFTR modulator and resultant hydration of viscous intestinal mucus, fecal matter may detach from the luminal wall and begin to move along the bowels simultaneously, increasing the potential for DIOS. Abdominal pain is a symptom of DIOS and can be for constipation as well [115]; therefore, abdominal pain reported in real-world studies, particularly cases that were severe and/or warranted interruption or discontinuation of therapy, may have been in relation to sudden increased fecal transit. More information, such as timing of onset, location and quality of abdominal pain, findings on physical assessment and imaging, would be necessary to evaluate this possibility.

Although CFTR is expressed in cervical epithelium and plays a role in hydration of cervical mucus throughout the menstrual cycle [116], this does not explain the menstrual abnormalities reported with LUM/IVA in real-world studies. In the phase 3 clinical trial setting, menstrual abnormalities reportedly occurred more frequently in female participants treated with LUM/IVA compared to placebo, with an even higher frequency in the subgroup of participants taking hormonal contraceptives [9]. This observation is very likely due to the induction of CYP3A and UDP-glucuronosyltransferase (UGT) enzymes by LUM, increasing the metabolism of and decreasing exposure to particular hormones in some hormonal contraceptive formulations [9]; this induction is not observed with other CFTR modulators, nor is the increased frequency of menstrual abnormalities. It was not specified whether the females in the real-world studies who experienced menstrual abnormalities were taking hormonal contraceptives, but this drug–drug interaction is a possible explanation.

Elevations in blood pressure were reported in the phase 3 clinical trials for LUM/IVA. At the end of the 96-week open-label extension of the TRAFFIC/TRANSPORT trials, study participants who continued on LUM/IVA 400 mg/250 mg every 12 h after or who rolled over to this regimen from placebo had mean absolute increases from baseline systolic blood pressure (SBP) of 5.9 mmHg and 5.1 mmHg, respectively, and mean absolute increases from baseline diastolic blood pressure (DBP) of 4.4 mmHg and 4.1 mmHg, respectively [113]. Moreover, one serious AE related to hypertension was reported in each of the TRAFFIC/TRANSPORT and PROGRESS trials [113]. A recent case report describes a six-year-old boy with CF whose intermittently-elevated blood pressure became persistently elevated (> 99th percentile for his age) after just two doses of LUM/IVA [117]. Upon further investigation, posaconazole, initiated about two weeks prior to the first documented incidence of hypertension, was deemed the likely culprit through inhibition of 11β-hydroxylase and resultant accumulation of mineralocorticoid precursors [117]. Interestingly, the individual in the real-world study who experienced a hypertensive emergency 12 h after his first LUM/IVA dose had been on posaconazole for five months prior, and, subjectively, his baseline blood pressure before LUM/IVA was relatively high (157/85 mmHg) [45]. Therefore, posaconazole and LUM independently may contribute to elevations in blood pressure, with more rapid elevations when used in combination. Elevations in blood pressure were also observed in the phase 3 study of ELX/TEZ/IVA, with mean absolute changes from baseline SBP of 3.1 mmHg and −0.1 mmHg in the ELX/TEZ/IVA and placebo groups at 24 weeks, respectively, and mean absolute changes from baseline DBP of 1.9 mmHg and 0.3 mmHg, respectively [23]. It has been hypothesized that people with CF have lower blood pressure when compared to age- and sex-matched controls due to salt wasting [118]; with increased CFTR function and diminished salt wasting, it may then be expected for elevations in blood pressure to occur with these agents. That said, the elevations in blood pressure are not seemingly a class effect and a greater increase in blood pressure was not observed with highly-effective CFTR modulators, as would have been expected. Single cases each of tachycardia and bradycardia were reported, but only the latter is reflective of other available information. Statistically significant decreases in heart rate were reportedly observed with LUM/IVA over placebo in the clinical trial setting, with a greater proportion of patients in the LUM/IVA group having heart rates < 50 beats per minute (bpm) [9]. Whether these changes were clinically significant cannot be determined from the information available, but the single case of bradycardia reported was asymptomatic [45]. In one study, people with CF were found to have a higher baseline heart rate when compared to healthy controls [119]; therefore, correction of CFTR function may theoretically reduce heart rate. However, again, this heart rate-lowering effect is seemingly unique to LUM/IVA.

CK elevations have been reported in clinical trials for all four CFTR modulators, with some cases being serious enough to warrant interruption or discontinuation [18,19,21,103,114]. The clinical significance of the CK elevations observed in both the clinical trials and real-world studies is unclear. In a large population-based study, 5.3% of individuals were found to have CK levels above the upper limit of normal established by age and sex; of those who obtained a “control” CK after three days of refraining from specific activities that may elevate CK (e.g., alcohol use, physical activity, and muscle training), 70% had normalization of CK values [120]. In four cases where the initial CK measures ranged from 5660 U/L to 15,941 U/L, the individuals had participated in significant physical activity and all had normalization of CK in their subsequent “control” test [120]. One clinical trial specified the CK elevations observed in both the placebo and ELX/TEZ/IVA groups were associated with exercise [23]; the frequency of CK elevations were 10% and 5% in the ELX/TEZ/IVA and placebo groups, respectively, perhaps reflecting that those receiving active therapy had greater capacity to be physically active. CFTR protein is expressed in peripheral muscular tissue and has been studied as a potential contributor to CF-related muscle abnormalities such as atrophy and weakness [121]; whether this too may explain some of the observed CK elevations with CFTR modulators is unclear.

An important consideration to keep in mind is the potential for erroneous attribution of AEs to medication; assignment of causality is subjective, and it is not uncommon for clinicians to inadvertently ascribe an AE to a medication in error, as is readily observed in placebo-controlled trials [122]. Moreover, while rare and unexpected AE may occur, a number of the AE reported in one or few individuals receiving CFTR modulators in the real-world studies did not have a clear pharmacokinetic or pharmacodynamic basis. Examples include dysphagia, pericarditis, worsening restless leg syndrome, swollen ear, tinnitus, secondary adrenal cortical insufficiency, and AML. Similar can be said for the single case of delayed onset first-degree heart block observed with concomitant TEZ/IVA and azithromycin reported [51]. Phase 3 clinical trials were significantly shorter than the upwards of eight months that it took for this drug interaction to be observed; however, available preliminary data for the open-label phase 3 rollover study for TEZ/IVA indicate this drug–drug interaction was not detected within the 100 weeks of total follow-up time [114]. Additionally important is the potential for reported AE to be secondary to CF itself, such as pulmonary exacerbation, hemoptysis, pneumothorax, and respiratory failure, especially in patients with lower levels of lung function.

### 4.2. Role of the CF Pharmacist in the Safe Prescribing of CFTR Modulators

As described, there are a number of potential AEs to be aware of with CFTR modulators. Although the safe initiation and monitoring of patients on CFTR modulators is a shared responsibility within a multidisciplinary CF team, pharmacists play an integral role. The following are highlights of pharmacists’ role in this, with additional suggestions and insights from the findings of this review.

#### 4.2.1. Selection of a CFTR Modulator

Depending on an individual’s *CFTR* genotype, there may be more than one CFTR modulator that he or she is eligible for. Efficacy outcomes aside, pharmacists can provide guidance into how the AE and drug–drug interaction profile of each CFTR modulator differs and may impact selection based on patient-specific factors. For example, LUM/IVA is not a favorable alternative in general for individuals who are homozygous for the *F508del* mutation with severe lung disease (i.e., ppFEV_1_ < 40%) and/or reactive airway disease due to the high frequency of respiratory-related AE and associated discontinuation. Moreover, due to the induction of CYP and UGT enzymes, LUM is prone to a myriad of drug–drug interactions [9,109]; pharmacists can help navigate which drug–drug interactions are clinically significant and warrant avoidance in favor of an alternative CFTR modulator when possible. Selection may be limited, however, when CFTR modulators are not accessible (e.g., due to lack of regulatory approval, or exclusion from formularies of eligible public or private healthcare benefits).

#### 4.2.2. Patient Counseling and Education

Before patients are initiated on CFTR modulators, pharmacists can provide the necessary counseling and education. Patients who are informed about potential AEs in advance may be more willing to trial a new medication [123,124], especially if strategies to address potential AEs are provided [123]. Anecdotally, this has been true at our site: some patients have expressed hesitancy and concern about potential AEs secondary to CFTR modulators, but once monitoring and strategies to address potential AE were discussed, they were more open to a trial. Similarly, some patients at our site were reluctant to obtain baseline and follow-up assessments, such as blood work or imaging, but were amenable once educated about the rationale (i.e., to ensure appropriate monitoring of both the safety and effectiveness of the medication).

It goes without saying that patients should be advised to inform their CF healthcare providers if new or changing symptoms arise, even if presenting as an expected AE, as it can be difficult to distinguish between something minor or self-limiting and something requiring timely attention. An example of this is increased respiratory secretions in the short-term following CFTR modulator initiation, often referred to as “the purge”. Though expected, increased respiratory secretions may also be manifest of a pulmonary exacerbation; the differentiation may be even more difficult with LUM/IVA due to additional respiratory AEs such as dyspnea, chest tightness, and declines in lung function.

Further salient is education regarding the potential for withdrawal syndrome upon discontinuation of CFTR modulators. There are several reports of patients experiencing clinical decline and/or pulmonary exacerbations upon discontinuation of a CFTR modulator, even requiring IV antibiotics in the community or hospital setting in some cases [27,125,126,127,128,129,130,131]. Therefore, patients should be informed to notify the CF healthcare providers if considering discontinuation of a CFTR modulator so that a plan to do so safely and with close monitoring can be established, and also to ensure refills are requested in advance to avoid lapses in therapy.

#### 4.2.3. Initiation and Monitoring Plan

Establishing patients’ baseline for parameters such as spirometry, blood work, vital signs, weight, and BMI, as well as clinical status for CF-related comorbidities is essential in monitoring for a number of the described AE following initiation of CFTR modulators. Pharmacists can also contribute to safe initiation and follow-up of CFTR modulators beyond these routine parameters.

Respiratory AE: The respiratory AE with LUM/IVA are well-established, but in cases where better-tolerated alternatives are not accessible, pharmacists can help establish an initiation plan to mitigate these AE. Insight and recommendations gleaned from the real-world studies are summarized in Table 4; although the majority of suggestions pertain to patients with severe lung disease, they may still benefit patients with higher lung function. Prior to LUM/IVA initiation, pharmacists can optimize patients’ pre-existing inhaled corticosteroid and bronchodilator therapies as well as establish a titration schedule. The titration schedule utilized by Murer et al. [31] was the most conservative, starting with just one tablet (one-quarter dose) and increasing to the target dose of two tablets twice daily over several days to weeks, depending on whether patients experienced AEs. Given that CFTR modulator therapy is intended to be long-term and LUM/IVA-related respiratory AE were predominantly reported in the initial weeks of therapy, a “start low, go slow” approach is reasonable. Further advisable is having patients in a closely-monitored clinic or inpatient setting at least for their first dose, with bronchodilators readily available. At our site, pre-treatment with a course of oral or IV antibiotics has also been a strategy to optimize lung health and minimize the potential for confounding in the assessment of a patient who does experience significant respiratory AE upon initiation of LUM/IVA. Walayat et al. [49] suggest patients may benefit from discontinuing hypertonic saline and continuing dornase alfa while on LUM/IVA to avoid patients “drowning” in liquefied secretions; although, a justifiable suggestion based on their case report, this may not be the experience of all individuals started on LUM/IVA or alternative CFTR modulators, and the risks versus benefits of discontinuing hypertonic saline (or dornase alfa) must be weighed against demonstrated benefit in pulmonary outcomes [132,133,134]. Whether it is safe to stop hypertonic saline or dornase alfa in patients on ELX/TEZ/IVA is currently under investigation [135], but unlike LUM/IVA, ELX/TEZ/IVA is considered a highly effective CFTR modulator and the study results would not be generalizable. Given that respiratory AEs have not been a notable concern with IVA, TEZ/IVA, and ELX/TEZ/IVA, there may be more comfort with initiating therapy at full-dose and in the community setting. That said, based on experience at our site and as described by Hebestreit et al. [30], when initiating patients with severe lung disease on highly effective CFTR modulators (i.e., IVA or ELX/TEZ/IVA) it is reasonable to consider an antibiotic “tune-up” and/or admission for close monitoring during the initial “purge”. We also recommend initiating ELX/TEZ/IVA at one orange tablet in the morning and one blue tablet in the evening for the first week in patients with severe lung disease to minimize the intensity of “the purge”, even if transitioning from LUM/IVA or TEZ/IVA.

DIOS Risk: Another potential reason our site would recommend a reduced dose of ELX/TEZ/IVA for the first week is the aforementioned concern for potential DIOS and other gastrointestinal-related AE, with the goal again being to reduce the intensity of effects at initiation. For the same reason, our site has included a bowel preparation (or abdominal X-ray to rule-out fecal loading) in the initiation protocol for ELX/TEZ/IVA; though not guided by published literature, it was felt the benefit of this intervention outweighed the risk. The pharmacist now educates patients regarding these gastrointestinal precautions and ensures completion of the preparation.

Drug–Drug Interactions with LUM: As previously mentioned, LUM is prone to drug–drug interactions due to induction of CYP and UGT enzymes; examples of medications impacted by this include azole antifungals, select psychotropic and antiepileptic medications (e.g., citalopram, sertraline, carbamazepine, and phenytoin), hormonal contraceptives, select immunosuppressants (e.g., tacrolimus and cyclosporine), and proton pump inhibitors [9,109]. How and whether to address a given interaction is not always straightforward. For individuals established on interacting medications prior to LUM/IVA initiation, depending on the medication and indication, close monitoring and adjusting the dose if necessary to maintain clinical stability may be a reasonable approach in some cases, whereas therapeutic drug monitoring (TDM) and close monitoring of drug levels before and after LUM/IVA initiation may be imperative in others. Although the pharmacist on the CF healthcare team may not be involved in the management of the medication(s) potentially impacted by LUM/IVA, he or she can liaise with the healthcare provider(s) responsible so that a plan may be implemented proactively.

Rash and Hypersensitivity: If a patient experiences a rash or other hypersensitivity reaction secondary to a CFTR modulator, pharmacists can work with the healthcare team to appropriately manage based on the severity and confirmed or suspected diagnosis. Anecdotally there are a myriad of approaches to managing patients who experience rash or hypersensitivity reactions, such as holding therapy until resolved then rechallenging (if appropriate), use of one or more medications (e.g., first- or second-generation antihistamines, topical corticosteroids, and systemic corticosteroids) until resolution, or, as described by Patterson et al. [28], developing and implementing a desensitization protocol. In females taking both ELX/TEZ/IVA and hormonal contraceptives, there are recommendations to consider interruption of both therapies should rash occur, and to consider stepwise resumption as tolerated following rash resolution [12]. Though these recommendations are founded on a clinical trial observation [23], the basis of the association is not clear or confirmed. Pharmacists may help evaluate the risks versus benefits of interrupting or discontinuing hormonal contraceptives in females who experience rash while on ELX/TEZ/IVA, and to identify suitable contraceptive alternatives, as necessary.

### 4.3. Limitations

The results of this review must be interpreted in the context of its limitations. Foremost, none of the studies had a comparator; however, even if there were a placebo or active comparator, it is possible, and not uncommon, for AEs to be misattributed to medications inadvertently [122]. It is possible that reported AEs were secondary to concomitant medications, comorbid conditions, the underlying CF itself, or other unrelated causes. Furthermore, validated methods to evaluate AEs were not consistently utilized. For example, in the case reports and case series describing changes in mental health and neuropsychiatric AEs following CFTR modulator initiation, in only one of the described cases was use of a validated tool (the GAD-7) reported [47,48,54]. This has the potential to overestimate AEs by misclassifying subjectively-reported symptoms, or underestimate AEs, as patients who do not self-report symptoms and are otherwise not evaluated regarding specific symptoms may go undetected. Similarly, safety was not a primary outcome for over one third of the studies available as full manuscripts; without systematic evaluation, it is possible AEs went undetected.

In addition to differences in the methodology for AE assessment, there was significant heterogeneity in both the size and characteristics of the patient populations across the included studies. As observed in this review and in the clinical trial setting, certain patient characteristics may be associated with a higher risk of particular AEs, and with smaller sample sizes, each individual AE results in a larger incremental change in the reported overall AE frequency. Therefore, the reported frequencies are not generalizable and must be interpreted in the context of the individual studies.

Although a robust search strategy was utilized in this systematic review, it is possible that relevant literature was missed. The majority of included studies were in the form of conference abstracts, which inherently do not include sufficient detail to assess the study quality or methodology for evaluating AEs. However, it was felt that exclusion of conference abstracts would risk missing potential signals for AEs that were not otherwise reported in the literature. Finally, a number of studies did not specify or clearly describe all of the AEs reported and/or did not specify the frequency of each AE within the study population, limiting the ability to more accurately characterize the CFTR modulator AE profiles.

## 5. Conclusions

As a growing number of people with CF gain access to CFTR modulators, it is prudent for healthcare providers to be aware of potential AEs as well as approaches for prevention and/or management to optimize patient safety. While the studies included in this review add value to the growing collective knowledge of potential AEs associated with CFTR modulators, this review also highlights the need for a systematic, comprehensive approach to monitoring for AEs in patients on these medications. It is also imperative that CF centers continue to share and publish their real-world experiences with patients on CFTR modulators to help better understand the potential short- and long-term AEs with these medications as well as patient characteristics that may be associated with higher risk of certain AEs. With this knowledge and their unique expertise within the CF healthcare team, pharmacists can play a key role in the safe initiation of people with CF on CFTR modulator therapy, as well as monitoring for and managing AEs that may arise thereafter.

## Figures and Tables

**Figure 1 jcm-10-00023-f001:**
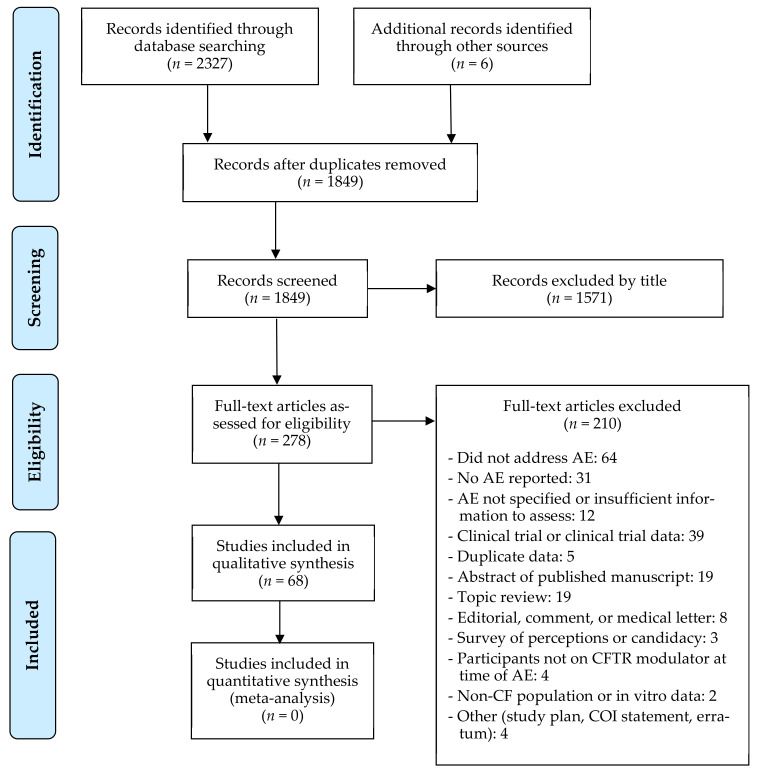
PRISMA flow diagram of study selection ^1^. ^1^ From: Moher, D.; Liberati, A.; Tetzlaff, J.; Altman, D.G.; The PRISMA Group. *P*referred *R*eporting *I*tems for *S*ystematic Reviews and *M*eta-*A*nalyses: The PRISMA Statement. *PLoS Med*. **2009**, *6*, e1000097, doi:10.1371/journal.pmed1000097.

**Table 1 jcm-10-00023-t001:** Summary of characteristics and results of cohort or survey studies with full manuscript.

Ref	Study Design & Location	Population ^a^	*n*	Recruitment Period & Follow-up Duration	Overall Adverse Events (AE) ^b,c^			Dose Modification, Interruption, or Discontinuation Due to AE ^b,c^		
**Ivacaftor**										
[29]	Prospective Cohort ^d^ United States	**Baseline Age** Pediatric and Adult - Mean: 33 yr - Range: 10–61 yr **CFTR Genotype** ≥ 1 copy G551D **Baseline ppFEV_1_** Mean: 30%	44	**Recruitment Period** Prior to commercial availability **Follow-up Duration** NS	**AE in *n* = 38 **(**86%**):	* n *	%	**Discontinuation:**	* n *	%
					- Pulmonary exacerbation - Hemoptysis - Increased sputum - Increased cough - URTI - Dyspnea - Abnormal respiration - Respiratory tract congestion - Headache - Rash	20 7 7 6 6 3 3 3 5 4	45 16 16 14 14 7 7 7 11 9	- Severe abdominal pain - Dizziness/tinnitus	1 1	2 2
					**SAE in *n* = 14 **(**32%**)**:**					
					- Pulmonary exacerbation - Hemoptysis - Pneumothorax - Acute respiratory failure - URTI - Abdominal pain - Gastroenteritis - Abnormal LFTs - Syncope - Secondary adrenocortical insufficiency	NS NS NS NS NS NS NS NS NS NS	- - - - - - - - - -			
[55]	Prospective Cohort United States Canada Italy	**Baseline Age** Pediatric and Adult - Mean: 17 yr - Range: 5–61 yr **CFTR Genotype** ≥1 gating mutation **Baseline ppFEV_1_** Mean: 86%	23	**Recruitment Period** Mar 2014 to Aug 2015 **Follow-up Duration** 3 mo	** 49 AE in *n* = 21 ** (**91%**) **:**	* n *	%	None reported		
					- Respiratory, unspecified - Gastrointestinal, unspecified - Infection, unspecified - Headache - Weakness - Dizziness - Fatigue	NS NS NS NS NS NS NS	- - - - - - -			
					** 5 SAE in *n* = 3 ** (**13%**) **:**	* n *	%			
					- Respiratory infection - Acute changes in metabolic and liver status	4 1	17 4			
[56]	Retrospective Cohort United Kingdom (1 center)	**Baseline Age** Pediatric - Mean: 9 yr - Range: 6–14 yr **CFTR Genotype** 1 copy G551D **Baseline ppFEV_1_** Mean: 68% ^e^	4	**Recruitment Period** Jan 2013 to Jun 2015 **Follow-up Duration** Mean: 24 mo	- Transaminitis (<3 x ULN)	*n* 1	% 25	None reported		
[57]	Retrospective Cohort Scotland (11 centers)	**Baseline Age** Pediatric - Median: 9 yr **CFTR Genotype** ≥1 copy G551D **Baseline ppFEV_1_** Mean: 85%	26	**Recruitment Period** NS (Jan 2013 to Mar 2013 for 85%) **Follow-up Duration** Mean: 17 mo	- Headache - Swollen ear - Cataracts	*n* 1 1 2	% 4 4 17 ^f^	None reported		
[58]	Retrospective Cohort France (25 centers)	**Baseline Age** Pediatric and Adult - Median: 18 yr - Range: 6–52 yr **CFTR Genotype** ≥1 copy G551D **Baseline ppFEV_1_** Mean: 72%	57	**Recruitment Period** Pre-1 Jun 2013 up to 30 Sep 2014 **Follow-up Duration** Up to 2 yr	** 34 AE in *n* = 21 ** (**37%**) **:**	* n *	%	** Interruption in *n* = 7 ** (**12%**) **:**	* n *	%
					- Transaminitis - Rhinopharyngitis - Asthma - Fever - Chest pain - Abdominal pain - Nausea or vomiting - Intestinal dysmotility - Headache - Fatigue - Rash or eczema - Depression - Myalgia - Arthritis - Breast hypertrophy - Orchitis - Atrial fibrillation	3 NS NS NS NS NS NS NS NS NS NS NS NS NS NS NS NS	5 - - - - - - - - - - - - - - - -	- Hepatitis - Rhinopharyngitis - Abdominal pain - Vomiting - Headache - Rash - Severe depression	NS NS NS NS NS NS NS	- - - - - - -
								**Discontinuation**:		
								- Transaminitis - Liver cirrhosis diagnosis	1 1	2 2
[30]	Retrospective Cohort ^d^ Germany (multicenter)	**Baseline Age** Adult - Mean: 34 yr **CFTR Genotype** ≥1 copy G551D **Baseline ppFEV_1_** Mean: 25%	14	**Recruitment Period** Sep 2012 to Apr 2013 **Follow-up Duration** Mean: 235 days	- Increased bronchial and nasal secretions - Headache - Worsening RLS - Abdominal pain - Hyperbilirubinemia (mild) - Transaminitis (<3 to 4x ULN)	*n* 3 1 1 1 1 1	% 21 7 7 7 7 7	**Discontinuation**: - Increased bronchial and nasal secretions * * Trial of reduced dose before discontinuation	*n* 1	% 7
**Lumacaftor/Ivacaftor**										
[41]	Prospective Cohort France (1 center)	**Baseline Age** Pediatric - Mean: 16 yr **CFTR Genotype** ∆F508/∆F508 **Baseline ppFEV_1_** Mean: 87%	32	**Recruitment Period** Mar 2016 to Dec 2016 **Follow-up Duration** 4 h post-first dose	- Acute drop in ppFEV_1_ - Wheeze	*n* 32 3	% 100 9	None reported		
[42]	Prospective Cohort France (47 centers)	**Baseline Age** Pediatric and Adult - Mean: 22 yr **CFTR Genotype** ∆F508/∆F508 **Baseline ppFEV_1_** Mean: 65%	845	**Recruitment Period** 1 Jan 2016 to 31 Dec 2016 **Follow-up Duration** 12 mo	** AE in *n* = 494 ** (**59%**) **:**	* n *	%	**Interruption:**	* n *	%
					- Respiratory - Digestive - Menstrual abnormality - Fatigue - Headache - CK > 5xULN - Transaminitis (> 3xULN)	316 181 53 37 19 20 5	37 21 6 4 2 2 0.6	- Respiratory - ‘Non-respiratory’ **Discontinuation**: Respiratory - Chest tightness/dyspnea - Bronchospasm - Increased cough/sputum - Hemoptysis - Pneumothorax Non-respiratory - Diarrhea, abdominal pain - CK >10xULN + myalgia - Fatigue - Headache - Depression - Metrorrhagia - Transaminitis (>6xULN) - Cutaneous rash - Tachycardia	16 8 *n* 38 24 9 2 1 18 5 5 4 4 3 2 1 1	2 1 % 5 3 1 0.2 0.1 2 0.6 0.6 0.5 0.5 0.4 0.2 0.1 0.1
[59]	Prospective Cohort United States (1 center)	**Baseline Age** Pediatric and Adult - Mean: 23 yr - Range: 12–48 yr **CFTR Genotype** ∆F508/∆F508 **Baseline ppFEV_1_** Mean: 70%	26	**Recruitment Period** NS **Follow-up Duration** 6 mo	See **Discontinuation**			**Discontinuation:** - Transaminitis - Unspecified AE	*n* 1 4	% 4 15
[31]	Prospective Cohort ^d^ Switzerland (1 center)	**Baseline Age** Adult - Mean NR **CFTR Genotype** ∆F508/∆F508 **Baseline ppFEV_1_** Median: 30%	20	**Recruitment Period** Jan 2016 to Jan 2017 **Follow-up Duration** 1 mo		* n *	%	**Reduced dose**:	* n *	%
					- Dyspnea - 3 h - 24 h - 1 mo - Chest tightness - 3 h - 24 h - 1 mo - Increased sputum - 3 h - 24 h - 1 mo - Pulmonary exacerbation - 1 mo	0 1 1 1 10 1 1 8 3 2	- 5 5 5 50 5 5 40 15 10	- Respiratory intolerance **Discontinuation:** - Chest tightness (at 24 h)	3 1	15 5
[32]	Prospective Cohort Australia (1 center)	**Baseline Age** Adult - Mean: 27 yr **CFTR Genotype** ∆F508/∆F508 **Baseline ppFEV_1_** Median: 36%	12	**Recruitment Period** Jan 2016 to Oct 2016 **Follow-up Duration** 1 mo		* n *	%	**Discontinuation:**	* n *	%
					- Acute drop in ppFEV_1_ - Respiratory AE overall - 4 h - 24 h - 1 mo - Dyspnea - 4 h - 24 h - 1 mo - Chest tightness - 4 h - 24 h - 1 mo - Increased sputum - 4 h - 24 h - 1 mo - Pulmonary exacerbation	12 5 10 8 2 6 7 4 8 5 0 2 1 6	100 42 83 67 17 50 58 33 67 42 - 17 8 50	- Chest tightness/dyspnea * * *n* = 2 discontinued after 1mo follow-up (5 wk and 9 wk)	3	25
[33]	Prospective Cohort France (11 centers)	**Baseline Age** Adult - Mean: 31 yr **CFTR Genotype** ∆F508/∆F508 **Baseline ppFEV_1_** Mean: 32%	53	**Recruitment Period** Jan 2016 to Jun 2016 **Follow-up Duration** 3 mo	** AE in *n* = 34 ** (**63%**) **:**	* n *	%	**Discontinuation:**	* n *	%
					- Abnormal respiration - Dyspnea - Increased cough - Abdominal pain, nausea, diarrhea, or vomiting - Fatigue - Rash - Pruritus - Breast tension	13 11 3 9 2 1 1 1	25 21 6 17 4 2 2 2	- Respiratory intolerance - Vomiting - Fatigue	13 1 1	25 2 2
[25]	Prospective Cohort ^d,g^ Australia (1 center)	**Baseline Age** Adult - Mean: 27 yr **CFTR Genotype** ∆F508/∆F508 **Baseline ppFEV_1_** Mean: 36%	10	**Recruitment Period** NS **Follow-up Duration** 52 wk	**AE in *n* = 6 **(**60%**)**:** - Chest tightness/dyspnea - Headache	*n* 6 2	% 60 20	None reported		
[43]	Retrospective Cohort Ireland (1 center)	**Baseline Age** Pediatric - Mean: 14 yr **CFTR Genotype** ∆F508/∆F508 **Baseline ppFEV_1_** Mean: 77%	15	**Recruitment Period** Sep 2016 to Aug 2017 **Follow-up Duration** NS	- Acute drop in ppFEV_1_ - Chest tightness - Increased sputum	*n* 14 2 2	% 93 13 13	None reported		
[44]	Retrospective Cohort United States (1 center)	**Baseline Age** Pediatric and Adult - Mean: 25 yr - Range: 12–59 yr **CFTR Genotype** ∆F508/∆F508 **Baseline ppFEV_1_** Mean: 67%	116	**Recruitment Period** NS **Follow-up Duration** Up to 11 mo	** AE in *n* = 46 ** (**40%**) **:**	* n *	%	**Reduced dose:**	* n *	%
					- Chest tightness - Dyspnea - Increased cough - Diarrhea - Nausea - Decreased appetite - Rash	23 12 10 5 3 2 2	20 10 9 4 3 2 2	- AE not specified **Discontinuation**: - Reasons not specified ^h^	10 20	9 17
[60]	Retrospective Cohort Greece (1 center)	**Baseline Age** Pediatric and Adult - Mean: 16 yr ^i^ - Range: 12–23 yr **CFTR Genotype** ∆F508/∆F508 **Baseline ppFEV_1_** Mean 92% ^i^	62	**Recruitment Period** Mar 2016 to Aug 2017 **Follow-up Duration** 12 mo	- Chest tightness	*n* 2	% 3	**Discontinuation:** - Transaminitis - Cataract	*n* 1 1	% 2 2
[34]	Retrospective Cohort ^d^ Spain (multicenter)	**Baseline Age** Pediatric and Adult - Mean: 27 yr - Range: 10–45 yr **CFTR Genotype** ∆F508/∆F508 **Baseline ppFEV_1_** Mean: 32%	20	**Recruitment Period** 2016 **Follow-up Duration** 6 mo	** AE in *n* = 15 ** (**75%**) **:**	* n *	%	**Discontinuation:**	* n *	%
					- Chest tightness - Dyspnea - Headache - Weight loss - ‘Sickness’ (not defined) - Asthenia - Abdominal pain - Transaminitis	9 8 5 5 3 3 2 2	45 40 25 25 15 15 10 10	- Decreased ppFEV_1_ - AE not specified	1 6	5 30
[45]	Retrospective Cohort Canada (1 center)	**Baseline Age** Adult - Median: 32 yr **CFTR Genotype** ∆F508/∆F508 **Baseline ppFEV_1_** Median: 40%	22	**Recruitment Period** Apr 2016 to Jun 2018 **Follow-up Duration** Median: 10 mo	** AE in *n* = 19 ** (**86%**) **:**	* n *	%	**Discontinuation:**	* n *	%
					- Chest tightness - Wheeze - Dyspnea - Increased sputum - Increased cough - Flu-like symptoms - Elevated blood pressure - Headache - Nausea - Elevated AST - Anxiety - Bradycardia - Pleuritic chest pain	14 4 3 3 2 1 5 4 2 1 1 1 1	64 18 14 14 9 5 23 18 9 5 5 5 5	- Respiratory symptoms - Asymptomatic hypertension - Symptomatic hypertension - Headache - Hypertensive emergency - Anxiety	3 2 1 1 1	14 9 5 5 5
[61]	Retrospective Cohort United States (1 center)	**Baseline Age** Adult - Mean NR **CFTR Genotype** ∆F508/∆F508 **Baseline ppFEV_1_** Mean NR	82	**Recruitment Period** Jul 2015 to Jun 2016 **Follow-up Duration** 12 mo	See **Discontinuation**			**Discontinuation**: Total overall: - Chest tightness * - Diarrhea ** - Abdominal pain - Nausea ** - Dysphagia - Elevated LFTs - Pericarditis - Allergic reaction ** - Suspected Stevens–Johnson syndrome * *n* = 3 also had significant drop in ppFEV_1_ ** *n* = 1 also discontinued due to chest tightness	*n* 17 11 2 1 1 1 1 1 1 1	% 21 13 2 1 1 1 1 1 1 1
[26]	Retrospective Cohort ^d,g^ Australia (7 centers)	**Baseline Age** Adult - Mean: 31 yr **CFTR Genotype** ∆F508/∆F508 **Baseline ppFEV_1_** Mean: 37%	72	**Recruitment Period** Nov 2015 to Mar 2017 **Follow-up Duration** 12 mo		* n *	%	**Discontinuation:**	* n *	%
					- Chest tightness/dyspnea - Increased sputum - Decrease in ppFEV_1_ - Headache - Fatigue - Nausea - Rash	40 4 2 2 5 1 2	56 6 3 3 7 1 3	- Chest tightness/dyspnea	22	31
[27]	Case Series (Survey) ^j^ International (31 centers)	**Baseline Age** Adult - Mean: 30 yr **CFTR Genotype** ∆F508/∆F508 **Baseline ppFEV_1_** Mean: 59%	26	**Recruitment Period** Questionnaire sent in 2018–2019 **Follow-up Duration** NS	- Pulmonary exacerbation - Post-partum acute myelocytic leukemia	*n* 1 1	% 4 4	**Discontinuation:** - Chest tightness	*n* 2	% 8

^a^ When adult and pediatric patients both included, age range reported when possible; ^b^ Rates not reported for all AE, as indicated by ‘NS’; ^c^ To avoid redundancy, if AE only reported in context of dose modification, interruption, and/or discontinuation of therapy, it was not listed in overall AE; ^d^ Study population part of a compassionate, ‘expanded access’, ‘managed access’, or ‘named patient’ program; ^e^ Mean calculated from *n* = 3 (75%) of study subjects, as baseline not reported for *n* = 1 (25%); ^f^ Frequency of 17% based on *n* = 12 screened; 8% frequency for overall cohort of *n* = 26; ^g^ Study was case-control, but only LUM/IVA-treated participants included in systematic review; therefore, assessed as cohort study; ^h^ Reason for discontinuation was not consistently assessed, and may include reasons unrelated to AE; ^i^ Mean baseline age and ppFEV_1_ based on *n* = 52 in final analysis of outcomes assessing effectiveness; *n* = 10 excluded from this analysis; ^j^ This case series is included in Table 1 due to results being presented in aggregate. **AST**, aspartate aminotransferase; **CFTR**, cystic fibrosis transmembrane conductance regulator; **CK**, creatine kinase; **h**, hour(s); **LFT**, liver function test; **mo**, month(s); **NR**, not reported; **NS**, not specified; **ppFEV_1_**, percent predicted Forced Expiratory Volume in 1 sec; **RLS**, restless leg syndrome; **SAE**, serious adverse events; **ULN**, upper limit of normal; **URTI**, upper respiratory tract infection; **wk**, week(s); **yr,** year(s).

**Table 2 jcm-10-00023-t002:** Summary of characteristics and results of cohort or survey studies in abstract form.

Ref	Study Design	Population	*n*	Overall Adverse Events (AE) ^a,b^			Dose Modification, Interruption, or Discontinuation Due to AE ^a,b^		
**Ivacaftor**									
[62]	Prospective Cohort	**Baseline Age** Pediatric - Mean: 5 yr **CFTR Genotype** ≥1 gating mutation **Baseline ppFEV_1_** Mean NR	4	**AE in *n* = 2 **(**50%**)**:** - URTI - Nasal congestion - Headache	*n* NS NS NS	% - - -	None reported		
[63]	Retrospective Cohort	**Baseline Age** Pediatric - Mean: 6 yr **CFTR Genotype** ≥1 gating mutation **Baseline ppFEV_1_** Median: 87%	10	- Transient rash - Increased obesity	*n* 2 1	% 20 10	None reported		
[64]	Prospective Cohort	**Baseline Age** Pediatric and Adult - Mean NR **CFTR Genotype** ≥1 copy S549R **Baseline ppFEV_1_** Mean: 54%	15	- Liver enzyme derangement	*n* 2	% 13	None reported		
[65]	Cross-sectional Survey	**Baseline Age** Adult - Mean: 26 yr **CFTR Genotype** ≥1G551D **Baseline ppFEV_1_** Mean: 62%	11 ^d^	**AE in *n* = 8 **(**73%**) **^d^****:** - Transient rash - Dizziness - Unspecified AE	*n* NS NS NS	% - - -	None reported		
**Lumacaftor/Ivacaftor**									
[66]	Prospective Cohort	**Baseline Age** Pediatric - Mean: 13 yr **CFTR Genotype** ∆F508/∆F508 **Baseline ppFEV_1_** Mean: 91%	14	- Acute drop in ppFEV_1_ (asymptomatic) - Chest tightness, tachypnea (requiring oxygen)	*n* 1 1	% 7 7	**Reduced dose ***: - Chest tightness, tachypnea * Eventual titration to full dose	*n* 1	% 7
[67]	Prospective Cohort	**Baseline Age** Pediatric - Mean: 14 yr **CFTR Genotype** ∆F508/∆F508 **Baseline ppFEV_1_** Mean: 87%	13	- Drop in ppFEV_1_ requiring salbutamol	*n* 7	% 54	None reported		
[68]	Prospective Cohort	**Baseline Age** Pediatric and Adult - Mean: 23 yr ^e^ **CFTR Genotype** ∆F508/∆F508 **Baseline ppFEV_1_** Mean 61% ^e^	369		* n *	%	**Discontinuation**:	* n *	%
				- Bronchospasm - Dyspnea - Abnormal respiration - Unspecified respiratory AE - Unspecified AE	15 12 7 4 120	4 3 2 1 33	- Unspecified AE	16	4
[69]	Prospective Cohort	**Baseline Age** Pediatric and Adult - Mean: 25 yr **CFTR Genotype** ∆F508/∆F508 **Baseline ppFEV_1_** Mean NR	311	**379 AE in *n* = 213 **(**68%**)**:** - Dyspnea - Cough - GI discomfort (e.g., diarrhea, nausea, abdominal pain) - Headache - Fatigue - Unspecified	n ^f^ NS NS NS NS NS NS	% 31 6 31 6 5 NR	**Interruption **(**stop/restart**)**:** - Unspecified AE and other reasons ^g^ **Discontinuation**: - Unspecified AE and other reasons ^g^	*n* 12 42	% 4 14
[35]	Prospective Cohort ^c^	**Baseline Age** Adult - Median: 31 yr **CFTR Genotype** ∆F508/∆F508 **Baseline ppFEV_1_** Median: 28%	14	- Chest tightness, breathless - Rash	*n* 7 1	% 50 7	**Discontinuation**: - Respiratory AE and/or rash	*n* 4	% 29
[70]	Prospective Cohort	**Baseline Age** Adult - Mean NR **CFTR Genotype** ∆F508/∆F508 **Baseline ppFEV_1_** Mean NR	29	- Chest tightness * * *n* = 4 cases severe, requiring hospitalization for IV steroids and antibiotics	*n* 13	% 45	**Reduced dose:** - Chest tightness **Discontinuation**: - Chest tightness	*n* 2 5	% 7 17
[36]	Prospective Cohort ^c^	**Baseline Age** Mean NR ^h^ **CFTR Genotype** ∆F508/∆F508 **Baseline ppFEV_1_** Mean NR	32	**AE in 88%:**	n ^f^	%	** Interruption ** (**stop/restart**) **:**	* n *	%
				- Respiratory AE - Drop in ppFEV_1_	NS NS	87 -	- Unspecified AEA **Discontinuation**: - Unspecified AE	1 8	3 25
[71]	Retrospective Cohort	**Baseline Age** Pediatric and Adult - Mean NR **CFTR Genotype** ∆F508/∆F508 **Baseline ppFEV_1_** Mean NR	34	**AE in *n* = 29 **(**85%**)**:** - Pulmonary exacerbation - Chest tightness - Dyspnea - Diarrhea - Abdominal pain	*n* 16 9 3 3 3	% 47 26 9 9 9	**Discontinuation:** - Unspecified AE	*n* 10	% 29
				** Serious AE in *n* = 8 ** (**24%**) **:**					
				- Respiratory failure ^i^ - Unspecified AE	1 7	3 21			
[72]	Retrospective Cohort	**Baseline Age** Pediatric and Adult - Mean: 26 yr **CFTR Genotype** ∆F508/∆F508 **Baseline ppFEV_1_** Mean: 68%	103	See **Discontinuation**			**Interruption/discontinuation ^j^**: - Chest tightness and/or pain - Elevated LFTs	*n* 17 NS	% 17 -
[73]	Retrospective Cohort	**Baseline Age** Adult - Mean: 31 yr **CFTR Genotype** ∆F508/∆F508 **Baseline ppFEV_1_** Mean: 50%	71	** AE in *n* = 41 ** (**58%**) **:**	* n *	%	**Discontinuation:**	* n *	%
				- Chest tightness - Dyspnea - Increased cough - GI (pain, constipation, or diarrhea) - Rash - Pruritus - Irregular menses or metrorrhagia - Breast tension - Headache - Myalgia	22 8 4 6 4 1 3 2 1 1	31 11 6 9 6 1 4 3 1 1	- Dyspnea - Chest tightness - Increased cough - Fatigue	7 6 3 1	10 9 4 1
[74]	Retrospective Cohort	**Baseline Age** Mean NR ^h^ **CFTR Genotype** ∆F508/∆F508 **Baseline ppFEV_1_** Mean NR	54	See **Discontinuation**			**Discontinuation**: - Chest tightness, dyspnea, and/or drop in ppFEV_1_	*n* 8	% 15
[75]	Retrospective Cohort	**Baseline Age** Adult - Mean: 31 yr **CFTR Genotype** ∆F508/∆F508 **Baseline ppFEV_1_** Mean NR	28	- Increased work of breathing or chest tightness - Drop in ppFEV_1_	*n* 12 5	% 43 18	**Discontinuation** - Respiratory intolerance vs. pulmonary exacerbation - Persistent respiratory intolerance/chest tightness - Rash and swelling of face - Increased anxiety	*n* 1 3 1 1	% 4 11 4 4
[76]	Retrospective Cohort	**Baseline Age** Adult - Mean NR **CFTR Genotype** ∆F508/∆F508 **Baseline ppFEV_1_** Mean NR	46		* n *	%	**Discontinuation:**	* n *	%
				- Drop in ppFEV_1_ - Transaminitis	21 2	46 4	- Dyspnea, cough, CFPEx, and/or chest tightness - Transaminitis - Headache - Muscle ache - Fatigue - Rash	4 1 NS NS NS NS	9 2 - - - -
[77]	Retrospective Cohort	**Baseline Age** Adult - Mean NR **CFTR Genotype** ∆F508/∆F508 **Baseline ppFEV_1_** Mean NR	28	See **Discontinuation**			**Discontinuation:** - SOB and/or drop in ppFEV_1_	*n* 15	% 54
[78]	Retrospective Cohort	**Baseline Age** Mean: 32 yr ^h^ **CFTR Genotype** ∆F508/∆F508 **Baseline ppFEV_1_** Mean: 62%	20	**Overall AE:**	* n *	%	** Interruption ** (**stop/restart**) **:**	* n *	%
				- Chest tightness - Elevated LFTs - Upset stomach - Increased stool output - Rash - Elevated thyroid function test - RA exacerbation	NS NS NS NS NS NS NS	- - - - - - -	- Unspecified AE - full-dose restart - half-dose restart **Discontinuation**: - Unspecified AE	2 4 2	10 20 10
[79]	Retrospective Cohort	**Baseline Age** Mean NR ^h^ **CFTR Genotype** ∆F508/∆F508 **Baseline ppFEV_1_** Mean NR	60	- Heartburn/reflux - Abdominal pain - Loose/oily stools	*n* 20 19 17	% 33 32 28	None reported		
[80]	Retrospective Cohort	**Baseline Age** Mean: 29 yr ^h,k^ **CFTR Genotype** ∆F508/∆F508 **Baseline ppFEV_1_** Mean: 80% ^k^	34	See **Discontinuation**			**Discontinuation**: Overall total: - Respiratory AE (70%) ^f^ - Unspecified reasons ^g^	*n* 11 NS NS	% 32 - -
[81]	Cohort ^l^	**Baseline Age** Pediatric - Mean NR **CFTR Genotype** ∆F508/∆F508 **Baseline ppFEV_1_** Mean NR	39	- AST >3x ULN	*n* 2	% 5	None reported		
[82]	Cohort ^l^	**Baseline Age** Pediatric and Adult - Range: 13–48 yr **CFTR Genotype** ∆F508/∆F508 **Baseline ppFEV_1_** Mean NR	47	See **Discontinuation**			**Discontinuation:** - Thoracic oppression and unspecified AE	*n* 4	% 9
[83]	Cohort ^l^	**Baseline Age** Adult - Mean: 28 yr **CFTR Genotype** ∆F508/∆F508 **Baseline ppFEV_1_** Mean: 61%	46	See **Discontinuation**			**Discontinuation:** - Dyspnea, increased sputum, and unspecified AE	*n* 6	% 13
[37]	Cohort ^c,l^	**Baseline Age** Adult - Mean: 31 yr **CFTR Genotype** ∆F508/∆F508 **Baseline ppFEV_1_** Mean: 28%	30	- Drop in ppFEV_1_ - Dyspnea, chest tightness, or chest pain - Increased sputum * Based on 31 trials of LUM/IVA in 30 patients	*n* 30 * 25 * NS	% 97 81 -	**Discontinuation:** - Respiratory AE, unspecified - Hypertension	*n* 3 1	% 10 3
[84]	Cohort ^l^	**Baseline Age** Adult - Mean: 31yr **CFTR Genotype** ∆F508/∆F508 **Baseline ppFEV_1_** Mean: 40%	8	See **Interruption**			**Interruption in *n* = 1 **(**13%**): - Drop in ppFEV_1_ - Eczema	*n* 1 1	% 13 13
[38]	Cohort ^c,l^	**Baseline Age** Mean NR ^h^ **CFTR Genotype** ∆F508/∆F508 **Baseline ppFEV_1_** Mean NR	19	See **Discontinuation**			**Discontinuation**: - Chest tightness and dyspnea	*n* 4	% 21
[85]	Cross-sectional questionnaire	**Baseline Age** Mean NR ^h^ **CFTR Genotype** ∆F508/∆F508 **Baseline ppFEV_1_** Mean NR	11	**AE in *n* = 5 **(**46%**)**:** - Increased cough - Chest pain - Trouble breathing - Chest tightness - Stomach pain	* n *	%	**Discontinuation:**	* n *	%
					4 2 2 1 1	36 18 18 9 9	- Increased cough	1	9
**Tezacaftor/Ivacaftor**									
[86]	Prospective Cohort	**Baseline Age** Pediatric - Mean: 16 yr **CFTR Genotype** ∆F508 homozygous or heterozygous **Baseline ppFEV_1_** Mean: 82%	72	See **Discontinuation**			**Discontinuation**: Overall total: - New-onset hemoptysis - Persistent nausea/vomiting - Elevated LFTs - Mental health changes - Alterations in blood glucose - Acholic stools	*n* 8 NS NS NS NS NS NS	% 11 - - - - - -
[87]	Prospective Cohort	**Baseline Age** Mean NR ^h^ **CFTR Genotype** NR **Baseline ppFEV_1_** Mean NR	50	- AE not specified	*n* 5	% 10	**Discontinuation:** - Liver function abnormalities	*n* 1	% 2
[88]	Prospective Cohort	**Baseline Age** Adult - Mean: 34 yr **CFTR Genotype** ∆F508/∆F508 **Baseline ppFEV_1_** Mean: 51%	5 ^m^	** AE in *n* = 5 ** (**11%**) **^m^** **:**	* n *	% ^m^	**Discontinuation:**	* n *	% ^m^
				- Sleep pattern disturbance - Out of body experience - Visual hallucination - Depersonalization - “Brain fog” - Severe migraine	2 1 1 1 1 1	5 2 2 2 2 2	- Out of body experience, visual hallucination - Depersonalization, “brain fog” - Severe migraine	1 1 1	2 2 2
[89]	Retrospective Cohort	**Baseline Age** Adult - Mean NR **CFTR Genotype** ∆F508/∆F508 **Baseline ppFEV_1_** Mean NR	18	See **Discontinuation**			**Discontinuation**: - Hair loss and fatigue	*n* 1	% 6
[39]	Cohort ^c,l^	**Baseline Age** Adult - Mean NR **CFTR Genotype** NR **Baseline ppFEV_1_** Mean: 34%	22	**AE in *n* = 3 **(**14%**)**:** - Rash - Blurred vision - Viral symptoms	*n* 2 1 1	% 9 5 5	**Discontinued:** - Blurred vision	*n* 1	% 5
**Elexacaftor/Tezacaftor/Ivacaftor**									
[40]	Retrospective Cohort	**Baseline Age** Adult - Mean: 36 yr **CFTR Genotype** ≥1 copy ∆F508 **Baseline ppFEV_1_** Mean: 31%	11	- Transaminitis	*n* 4	% 36	None reported		

^a^ Rates not reported for all AE, as indicated by ‘NS’; ^b^ To avoid redundancy, if AE only reported in context of dose modification, interruption, and/or discontinuation of therapy, it was not listed in overall AE; ^c^ Study population part of a compassionate, ‘early access’, ‘expanded access’, ‘managed access’, or ‘named patient’ program; ^d^ Total study cohort of *n* = 11, but only *n* = 9 patients completed symptom questionnaire; AE frequency calculated based on total *n* = 11, ^e^ Mean baseline age and ppFEV_1_ based on *n* = 135 in final analysis of outcomes assessing effectiveness; *n* = 234 excluded from this analysis; ^f^ As reported, unable to accurately determine the absolute number of patients who experienced AE ; ^g^ Frequency of specific reasons for interruption or discontinuation not clear and include reasons unrelated to AE; ^h^ Included age groups (i.e., pediatric and/or adult) not specified; ^i^ Respiratory failure occurred in 1 individual on two occasions, both within 24 h of initiating and reinitiating LUM/IVA; ^j^ Of the *n* = 25 who stopped, 9 restarted and 6 of experienced the same AE; unclear which AE the 3 who restarted experienced and whether the 6 who experienced the same AE then discontinued permanently; ^k^ Mean baseline age and ppFEV_1_ based on *n* = 23 in final analysis of outcomes assessing effectiveness; *n* = 11 excluded from this analysis; ^l^ Unable to discern if prospective versus retrospective based on reported information; ^m^ Total study cohort of *n* = 44, but focused on neurocognitive AE in *n* = 5; AE frequencies calculated based on total *n* = 44. **AST**, aspartate aminotransferase; **CFTR**, cystic fibrosis transmembrane conductance regulator; **CFPEx**, cystic fibrosis pulmonary exacerbation; **GI**, gastrointestinal; **LFT**, liver function test; **LUM/IVA**, lumacaftor/ivacaftor; **NR**, not reported; **NS**, not specified; **ppFEV_1_,** percent predicted forced expiratory volume in 1 sec; **RA**, rheumatoid arthritis; **SOB**, shortness of breath; **ULN**, upper limit of normal; **URTI**, upper respiratory tract infection; **yr,** year(s).

**Table 3 jcm-10-00023-t003:** Summary of characteristics and results of case series and case reports.

Ref	Location	*n*	Patient Information	Description of Adverse Events (AE) ^a^	Dose Modification, Interruption or Discontinuation of Therapy, or Other Intervention(s) Due to AE
**Ivacaftor**					
[46]	Germany	2	**Case 1**: 55F **Case 2:** 55F	– Bronchial tightness – AST and ALT 2-3x ULN	– No interventions reported – resolved by approximately 14 days – **IVA decreased to half-dose**; transaminase subsequently normalized
**Lumacaftor/Ivacaftor**					
[47]	United States	5	**Case 1**: 15F	– 2 wk: new depression and SI – Within 7 mo: increased anxiety attack	– Initiation of CBT and sertraline – **LUM/IVA discontinued at 11 mo** (anxiety, menstrual irregularity); mood improved within 10 days of discontinuation
			**Case 2**: 17F	– Within 2 mo: new worsening baseline depression, new SI	– Initiation of CBT, increased fluoxetine dose – **LUM/IVA discontinued at 9 mo** (GI intolerance); mood improved following discontinuation
			**Case 3**: 14F	– Within 9 mo: new depression and SI – Suicide attempt (ibuprofen overdose)	– Initiation of CBT and fluoxetine; psychiatric hospitalization – **LUM/IVA discontinued at 12 mo** (depression, SI); mood improved following discontinuation
			**Case 4:** 12F	– 2 mo: new depression and SI – Suicide attempt (cutting)	– Initiation of CBT; psychiatric hospitalization; After LUM/IVA discontinuation: initiation of sertraline – **LUM/IVA discontinued at 3 mo** (depression, SI); mood improved within 3 wk of discontinuation
			**Case 5:** 17F	– 7 mo: worsening baseline depression – After LUM/IVA discontinuation: suicide attempt (escitalopram overdose)	– Initiation of escitalopram; After LUM/IVA discontinuation: trials of mirtazapine and fluoxetine; psychiatric hospitalization – **LUM/IVA discontinued at 7 mo** (depression) – Continued worsening depression; **LUM/IVA restarted months later**
[48]	United States	4	**Case 1**: 44F	– Within 1 mo: worsening baseline depression and anxiety, limiting adherence to CF therapies – Increased cough and sputum, ppFEV_1_ decline from 74% to 49%	– Increased citalopram dose; citalopram replaced by alternative psychotropic medications; hospitalizations for CFPEx management – **LUM/IVA discontinued at 4 mo** (depression, anxiety); mood and anxiety improved, but not to baseline until prior citalopram resumed
			**Case 2**: 26M	– Within 3 mo: worsened depression (baseline bipolar disorder and recent bereavement) – ppFEV_1_ decline from 69% to 44%	– Change in psychotropic medications; CFPEx management – **LUM/IVA continued unchanged** – Mood and ppFEV_1_ back to baseline several months after interventions
			**Case 3**: 36M	– Worsening baseline anxiety and opioid use disorder; subsequent worsening baseline depression, limiting medication adherence – ppFEV_1_ decline from 38% to 28%	– Change in psychotropic medications; hospitalization for psychiatric evaluation and CFPEx management; intensive outpatient mental health program – **LUM/IVA discontinued** (non-adherence)
			**Case 4**: 13M	– 9 mo: worsening baseline anxiety (supported by increased GAD-7 score) with clinical impairment	– Re-initiation of CBT; increased citalopram dose – **LUM/IVA continued unchanged**; ongoing residual anxiety
[49]	United States	1	**Case**: 31F	– Within 6 wk: increased fatigue, “more winded”, dyspnea, “drowning” in sputum and sinus secretions – ppFEV_1_ decline from 58% to 34%	– Hospitalization for CFPEx management and aggressive pulmonary rehab – **LUM/IVA continued unchanged**; respiratory symptoms ‘near’ baseline and ppFEV_1_ improved to 42% (below baseline) after 2-wk hospitalization
[90]	United States	1	**Case**: 36M	– Immediate severe dyspnea, chest tightness – ppFEV_1_ drop from 31% to 23%; SpO_2_ drop from 95% at rest to 87% on exertion	– **LUM/IVA decreased to half-dose** (respiratory AE) – **LUM/IVA discontinued at 6 mo** (ongoing respiratory AE despite dose reduction); immediate improvement in respiratory AE, resolution of hypoxia on exertion, 1 mo ppFEV_1_ returned to baseline
[50]	Germany	1	**Case**: 20F	– 2 wk: malaise, severe progressive rash, pruritus, and facial swelling – Skin prick test negative for acute or delayed reaction to IVA, LUM/IVA, and TEZ/IVA – In vitro T-cell-mediated reaction to LUM	– Hospitalization for high-dose corticosteroids – **LUM/IVA discontinued**; rapid resolution of symptoms following discontinuation
[28]	United States	1	**Case**: 18F	– 7 days: non-confluent, red, bumpy, pruritic rash on shins and forearms; persisted 1 mo – Symptom recurrence upon rechallenging (within 3 and 7 days of first and second rechallenge, respectively)	– **LUM/IVA discontinued;** resolution of rash – **Retrialed at full- and half-dose** (retrials both 1 wk after discontinuation) – **LUM/IVA discontinued indefinitely** (recurrent rash); resolution of rash following discontinuation
[91]	United States	1	**Case**: 27M	– 4 wk: abdominal pain, diarrhea, N/V – AlkP 8477 U/L (other LFTs WNL)	– Hospitalization for supportive care, rule-out other cause – **LUM/IVA discontinued**; AlkP declined to WNL by 6 mo later
[92]	France	1	**Case**: 26M	– New myalgia following acyclovir initiation – Weeks later: rhabdomyolysis with worsening myalgia, muscle edema, dark urine, CK 17,582 U/L, AST 10x ULN, and ALT 4x ULN	– Hospitalization for supportive care – **LUM/IVA discontinued** (along with acyclovir); CK returned to normal, but residual lower limb myalgia
**Tezacaftor/Ivacaftor**					
[28]	United States	1	**Case**: 18F	– 3 days: non-confluent, red, bumpy, pruritic rash on shins and forearms – Symptom recurrence within 7 days upon rechallenging	– **TEZ/IVA discontinued**; resolution of rash – **Retrialed 2 mo later**, **then discontinued** (recurrent rash) – IVA desensitization protocol, titrating to full-dose IVA over 10 days; **TEZ/IVA restarted successfully thereafter**
[51]	Australia	1	**Case**: 21F	– Long-term azithromycin therapy and no past cardiac arrhythmia before initiation – 36 wk: PR interval of 334 ms (WNL at baseline then 4, 12, and 24 wk). QT and QTc normal – After TEZ/IVA held, PR interval WNL at 4 wk – After TEZ/IVA retrialed: 39 wk, PR interval 25–30 ms longer than at 4, 8, 12, and 24 wk	– Rule-out other causes (e.g., echocardiogram, blood work, viral testing) – **TEZ/IVA held** (PR prolonged on repeat ECG at 37 and 38 wk) – **TEZ/IVA retrialed ~8 mo later** – Azithromycin discontinued at 39 wk (PR interval lengthening on ECG); PR interval back to baseline 4 wk later and remained normal thereafter – **TEZ/IVA continued**
**Elexacaftor/Tezacaftor/Ivacaftor**					
[52]	United States	7	**Case 1**: 23M	– 12 days: new right testicular discomfort, intermittent sharp pain, sensitivity to touch and pressure	– OTC analgesics – **ELX/TEZ/IVA unchanged**; symptoms resolved in 12 days
			**Case 2**: 23M	– 3 days: new right testicular soreness	– OTC analgesics – **ELX/TEZ/IVA unchanged**; symptoms resolved in 7 days
			**Case 3**: 17M	– 2 days: new left testicular pain, difficulty urinating, lower abdominal pain	– No interventions reported – **ELX/TEZ/IVA unchanged**; symptoms resoled in 1 day
			**Case 4**: 22M	– 7 days: new sharp, twisting, cramping testicular pain	– **ELX/TEZ/IVA interrupted for 2 days, restarted at full dose**; symptoms resolved in 7 days, no recurrence reported
			**Case 5**: 30M	– 3–4 days: new right testicular pain and discomfort, increased ejaculate volume – Constipation identified on abdominal CT	– No interventions reported – **ELX/TEZ/IVA unchanged**; symptoms resolved in 21 days
			**Case 6**: 37M	– 2–3 days: new testicular pain and discomfort when driving “bumpy roads”	– No interventions reported – **ELX/TEZ/IVA unchanged**; symptoms resolved in 2–3 days
			**Case 7**: 39M	– 3 days: new left testicular pain – PE and ultrasound findings of bilateral epididymoorchitis and scrotal wall cellulitis	– OTC analgesics; antibiotic for epididymoorchitis – **ELX/TEZ/IVA dose reduced to 1 tab daily**, **titrated to full dose**; symptoms resolved in 1–2 days, no recurrence reported
[53]	United States	7	**Case 1**: 38F	– Within 1 mo: RUQ pain – Peak LFTs: AST 79 U/L, ALT 59 U/L, AlkP 103 U/L, total bili 12 umol/L	– Laparoscopic cholecystectomy (chronic cholecystitis with cholelithiasis; serosal fibrous adhesions) ^b^ – **ELX/TEZ/IVA unchanged** (except morning dose held pre-op)
			**Case 2**: 33F	– Day 1: RUQ pain, nausea – Peak LFTs: AST 69 U/L, ALT 106 U/L, AlkP 138 U/L, total bili 55 umol/L	– Laparoscopic cholecystectomy (acute cholecystitis with cholelithiasis, mucosal necrosis) ^b^; common biliary duct stent for persistent bile leak – **ELX/TEZ/IVA held Day 3–7** (biliary colic), restarted post-op; **held 1.5 days peri-op for stent placement** (post-op complications) ^c^ – **Resumed full dose post-stent placement**; symptoms resolved
			**Case 3**: 28M	– Day 3: epigastric radiating to RUQ; N/V – Peak LFTs: AST 34 U/L, ALT 37 U/L, AlkP 259 U/L, total bili 9 umol/L	– Laparoscopic cholecystectomy (chronic cholecystitis with cholelithiasis, extensive mucosal erosion, serositis, wall fibrosis) ^b^ – **ELX/TEZ/IVA unchanged** (except morning dose held pre-op)
			**Case 4**: 28M	– Within 1 day: RUQ pain, nausea ^c^ – Peak LFTs: AST 42 U/L, ALT 65 U/L, AlkP 121 U/L, total bili 22 umol/L	– Laparoscopic cholecystectomy (chronic cholecystitis with cholelithiasis) ^b^ – **ELX/TEZ/IVA held 2 wk pre-op, resumed 4 wk post-op**, symptoms resolved – **ELX/TEZ/IVA subsequently held** (transaminitis), plans to retrial reduced dose upon transaminase normalization ^c^
			**Case 5**: 40F	– Day 1: lower abdominal and epigastric pain – Day 7: severe RUQ pain – Peak LFTs: AST 49 U/L, ALT 56 U/L, AlkP 335 U/L, total bili 10 umol/L	– No surgical intervention – **ELX/TEZ/IVA unchanged**
			**Case 6**: 26F	– 2 wk: progressive abdominal pain; N/V – Peak LFTs: AST 516 U/L, ALT 283 U/L, AlkP 134 U/L, total bili 26 umol/L	– Laparoscopic cholecystectomy (chronic cholecystitis with cholelithiasis) ^b^ – **ELX/TEZ/IVA unchanged** (except morning dose held pre-op)
			**Case 7**: 27F	– Day 1: central abdominal pain, anorexia, N/V – Peak LFTs of AST 103 U/L, ALT 36 U/L, AlkP 253 U/L, total bili 10 umol/L	– Hospitalization for symptom management; elective laparoscopic cholecystectomy at later date (chronic cholecystitis with cholelithiasis) ^b^ – **ELX/TEZ/IVA held Day 5** (biliary colic), restarted Day 6 ^c^ – **ELX/TEZ/IVA held Day 7** (symptom recurrence), restarted Day 8 ^c^ – Symptoms resolved (only held morning dose pre-op thereafter) ^c^
[54]	United States	1	**Case**: 19F	– Within 2 wk: worsening baseline depression with new SI. At 3 wk, sleep paralysis with vivid hypnopompic hallucinations on 3 consecutive nights (history of 3 episodes of sleep paralysis previously) – After ELX/TEZ/IVA restarted: worsening depression and anxiety; subsequent improved depression and anxiety, resolved SI and sleep paralysis/hallucinations – 1–2 mo after full-dose ELX/TEZ/IVA: oscillating anxiety and depression, intermittent SI and sleep paralysis – After ELX/TEZ/IVA and IVA dose times switched: worsening depression and anxiety – After decrease to ELX/TEZ/IVA 2 tabs in AM: some improvement in depression; recurrence of sleep paralysis with hypnopompic hallucination	– **ELX/TEZA/IVA discontinued at ~3 wk** (worsening mood, sleep paralysis); resolution of sleep paralysis and hypnopompic hallucinations within 2 days – Sertraline initiated for ongoing depression. Subsequently, sertraline and baseline quetiapine doses increased, clonazepam initiated – **ELX/TEZA/IVA restarted ~1 mo after discontinuation** (respiratory status worsened); **titrated to 1.5 ELX/TEZA/IVA tabs + 1.5 IVA tabs in AM** – Sertraline and clonazepam doses increased, quetiapine dose decreased, bupropion initiated adjunctively (bupropion later stopped due to intolerance) – **ELX/TEZA/IVA titrated to full-dose** over ~1 mo; **ELX/TEZ/IVA tabs moved to PM, IVA tab to AM** 1–2 mo later (worsening psychiatric symptoms) – **ELX/TEZA/IVA dose decreased to 2 ELX/TEZ/IVA tabs in AM** ~2 mo later (worsening depression and anxiety); improvement in depressive symptoms and cognition, sertraline, quetiapine, and clonazepam doses increased – **Continued 2 ELX/TEZ/IVA tabs in AM**; symptoms of depression, anxiety, and sleep paralysis remained above baseline

^a^ Time indicates onset from start of CFTR modulator; ^b^ Per the cholecystectomy pathology report; ^c^ Per personal communication with corresponding author **AlkP**, alkaline phosphatase; **ALT**, alanine aminotransferase; **AM**, ante meridiem (i.e., morning); **AST**, aspartate aminotransferase; **bili**, bilirubin; **CBT**, cognitive behavioral therapy; **CF**, cystic fibrosis; **CFPEx**, CF pulmonary exacerbation; **CFTR**, cystic fibrosis transmembrane conductance regulator; **CK**, creatine kinase; **CT**, computerized tomography; **ECG**, electrocardiogram; **ELX/TEZ/IVA**, elexacaftor/tezacaftor/ivacaftor; **IVA**, ivacaftor; **GAD-7**, Generalized Anxiety Disorder 7-item scale; **GI**, gastrointestinal; **LUM/IVA**, lumacaftor/ivacaftor; **mo**, month(s); **N/V**, nausea and vomiting; **OTC**, over-the-counter; **ppFEV_1_**, percent predicted Forced Expiratory Volume in 1 sec; **peri-op**, perioperatively; **PM**, post meridiem (i.e., evening); **pre-op**, pre-operatively; **RUQ**, right upper quadrant; **SI** suicidal ideation; **SpO_2_**, oxygen saturation; **TEZ/IVA**, tezacaftor/ivacaftor; **ULN**, upper limit of normal; **wk**, week(s); **WNL**, within normal limits.

**Table 4 jcm-10-00023-t004:** Study protocols, strategies, or suggested management and monitoring for reported adverse events.

CFTR Modulator	Ref	Adverse Event (AE)	Study Protocol/Strategy or Suggested Management/Monitoring for AE
**Protocols/Strategies utilized in study**			
LUM/IVA	[43]	Respiratory AE and acute drop in ppFEV_1_	**Initiation protocol:** Nebulized salbutamol 15 min prior to first dose Initiate half-dose (LUM 200 mg/IVA 125 mg twice daily), unless using granule formulation for pediatrics, which cannot be halved At 1 week, increase to full-dose (LUM 400 mg/IVA 250 mg twice daily) as tolerated
LUM/IVA	[31]	Respiratory AE resulting in discontinuation	**Initiation protocol:** First dose (1 tablet; LUM 200 mg/IVA 125 mg) in outpatient clinic with 3-h monitoring post-dose Clinic to follow-up with patient every third day to reassess: - increase dose by 1 tablet if tolerated - decrease dose by 1 tablet if AE If dose decreased due to AE: - once AE resolved and stable 2 weeks, increase dose by 1 tablet - if AE symptoms persist, decrease dose further by 1 tablet Goal to increase step-wise to full-dose (LUM 400 mg/IVA 250 mg q12 h)
LUM/IVA	[44]	Respiratory AE resulting in discontinuation	**Initiation strategy:** For patients with ppFEV_1_ ≤ 40%, make effort to ensure use of combination inhaled beta_2_-agonist and corticosteroid, and lower initial dose for ≥7 days before increasing to recommended full dose.
TEZ/IVA and IVA ^a^	[28]	Morbilliform drug eruption (Type IV drug allergy), recurrent upon rechallenge	**Desensitization protocol:** Prepare IVA dilutions as directed and follow 10-day desensitization protocol, with IVA doses escalating from 5 mcg to 150 mg Once IVA dose of 150 mg tolerated, continue 150 mg q12 h If plan to start TEZ/IVA, continue IVA 150 mg q12 h for 3 days before transition to TEZ/IVA If tolerated, counsel patient that ongoing adherence is prudent to avoid potential symptom recurrence and need for repeat desensitization
**Suggested Strategy, Management, and/or Monitoring as a Result of Observations**			
IVA	[30]	Difficulty clearing liquefied airway secretions	For patients with severe lung disease, consider hospitalization for IV antibiotics and intense physiotherapy prior to initiation and for the first week of therapy.
IVA ^b^	[57]	Cataract formation	In pediatric patients, baseline and follow-up ophthalmological examinations should be performed to assess for cataract formation.
LUM/IVA	[41]	Respiratory AE and acute drop in ppFEV_1_	Closely monitor patients in clinic in the hours following the first dose, especially if low baseline ppFEV_1_ and/or known reversible airway obstruction ≥ 12%.
LUM/IVA	[32]	Respiratory AE and acute drop in ppFEV_1_	For patients with ppFEV_1_ < 40%, consider lower initiation dose and monitor closely following initiation for lung function decline and respiratory AE.
LUM/IVA	[25]	Respiratory AE resulting in discontinuation	In patients with ppFEV_1_ < 40%, use caution with administration in case of poor tolerance.
LUM/IVA	[33]	Respiratory AE resulting in discontinuation	Pre-treat with long-acting bronchodilator therapy before initiation, as a strategy to mitigate respiratory AEs and potential subsequent discontinuation.
LUM/IVA	[61]	Respiratory AE resulting in discontinuation	Consider slow titration of dose during treatment initiation to mitigate AE.
LUM/IVA	[75]	Respiratory AE resulting in discontinuation	Consider slower titration of dose or increased use of bronchodilators to mitigate AE on initiation, especially in patients with ppFEV_1_ ≤ 40%.
LUM/IVA	[26]	Respiratory AE resulting in discontinuation	**Suggested consideration for patients homozygous for *F508del* mutation:** Preference for TEZ/IVA over LUM/IVA may be considered in patients with ppFEV_1_ < 40%, as TEZ/IVA does not share the same respiratory AE profile of chest tightness and dyspnea causing high rates of discontinuation.
LUM/IVA	[49]	Respiratory AE upon initiation, presenting as pulmonary exacerbation	Educate patients regarding potential for symptoms such as dyspnea, increased cough and sputum production, and malaise, as well as the importance of continuing pulmonary rehab and airway clearance Consider avoiding combination of both dornase alfa and hypertonic saline upon LUM/IVA initiation to minimize excess watery secretions and feeling of “drowning” in secretions
LUM/IVA	[45]	Hypertension	Monitor blood pressure routinely both short- and long-term, and manage accordingly.
LUM/IVA	[47]	New or worsening depression and anxiety	Monitor all patients routinely for new or worsening depression and anxiety In patients already on psychotropic medications, evaluate drug–drug interactions prior to LUM/IVA initiation and consider increased monitoring and potential need to adjust psychotropic medications’ dose to maintain clinical effect after initiation
LUM/IVA	[48]	Worsening depression and anxiety	**Suggested initiation monitoring and management in patients already on citalopram, escitalopram, or sertraline:** Close monitoring of mental health, only changing treatment plan if needed Counsel patients regarding potential for changes in mood and anxiety Complete PHQ-9 and GAD-7 scales before and after LUM/IVA initiation If available, consider drawing drug levels before and after LUM/IVA initiation, bearing in mind that changes in SSRI levels may not be clinically significant and the SSRI should be dosed to response and tolerability Any decisions to adjust psychotropic therapy and/or to decrease or stop LUM/IVA should be a shared decision with the patient and should consider potential for medical and/or psychiatric decline
LUM/IVA	[79]	Breakthrough heartburn or GERD due to increased PPI metabolism	**Suggested initiation monitoring and management for patients already on a PPI**: Maintain patients on the lowest effective dose, increasing PPI dose only if patients present with breakthrough symptoms of heartburn/GERD.
TEZ/IVA	[88]	Neurocognitive side effects	Counsel patients regarding the possibility of neurocognitive AE (e.g., sleep pattern disturbances, visual hallucinations, out-of-body experiences) when initiating.
TEZ/IVA and ELX/TEZ/IVA ^c^	[51]	First-degree heart block in combination with azithromycin	**Suggested monitoring for patients already on azithromycin** (**or initiating azithromycin in patients already on TEZ/IVA or ELX/TEZ/IVA ^b^**): Obtain ECG at baseline and 3, 6, 9, and 12 months after combination to monitor for PR interval prolongation (onset may be delayed >8 months).
ELX/TEZ/IVA	[52]	Testicular pain following initiation	Until more is known about its impact on the reproductive system and fertility, may consider advising male patients to use contraception if having intercourse with females of child-bearing age, or refer to appropriate family planning resources if interested in pursuing pregnancy.
ELX/TEZ/IVA	[53]	Biliary colic following initiation	Prior to initiation, assess for history of postprandial RUQ pain and/or biliary colic. Following initiation, closely monitor LFTs and symptoms of biliary colic.
ELX/TEZ/IVA	[54]	New or worsening depression, SI, anxiety, and/or sleep paralysis with hypnopompic hallucinations	**Suggested initiation monitoring**: Following initiation, screen and closely monitor for new and/or worsening symptoms of depression, SI, and anxiety hallucinations.

^a^ In the case reported, the patient had the morbilliform drug eruption while taking both LUM/IVA and TEZ/IVA; however, the described desensitization protocol was specific to IVA, with transition to TEZ/IVA; ^b^ In the observational study, patients were receiving IVA. However, the product monographs of LUM/IVA, TEZ/IVA, and ELX/TEZ/IVA also recommend baseline and follow-up ophthalmological examinations in pediatric patients initiating therapy; ^c^ In the case reported, the patient had PR interval prolongation with the combination of azithromycin and TEZ/IVA; therefore, it is suggested that the same may occur with azithromycin in combination with ELX/TEZ/IVA. **CFTR**, cystic fibrosis transmembrane conductance regulator; **ECG**, electrocardiogram; **ELX/TEZ/IVA**, elexacaftor/tezacaftor/ivacaftor; **GAD-7**, general anxiety disorder-7; **GERD**, gastroesophageal reflux disease; **h**, hour(s); **IVA**, ivacaftor; **LFTs**, liver function tests; **LUM/IVA**, lumacaftor/ivacaftor; **PHQ-9**, patient health questionnaire-9; **ppFEV_1_**, percent predicted Forced Expiratory Volume in 1 sec; **PPI**, proton pump inhibitor; **q12 h**, every 12 h; **RUQ**, right upper quadrant; **SI**, suicidal ideation; **SSRI**, selective serotonin reuptake inhibitor; **TEZ/IVA**, tezacaftor/ivacaftor.

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
