# Peer review of "Real-World Safety of CFTR Modulators in the Treatment of Cystic Fibrosis: A Systematic Review"

_jcm, 2020, doi:10.3390/jcm10010023_

Round 1

Reviewer 1 Report

This is a remarkably well-written submission. I found it to be both incredibly detailed and cautiously balanced.I suspect that it will be of tremendous use to the CF Care community. 

I have one global comment and then a few minor suggestions.

The global comment is that I think the authors might want to consider the distinction between when someone is a "patient with CF" as opposed to when he or she is a "person with CF", as this has become something of a cause in the CF community.  When someone is admitted to the hospital they are clearly a patient, but is everyone out in the world taking a chronic medication or with a chronic condition always a patient?  Its not common to refer to people on statins as cardiac patients or pregnant women as obstetric patients, although they are presumably under the care of a physician... most of us are.   Perhaps the submission could be read in this context to identify where "patients" can become "people" (e.g., line 781: "As a growing number of patients with CF gain access to CFTR modulators").

Minor points:

Lines 287-288. The clause "In 1 study that specified,..." seems awkward. I think I understand the point being made (that other studies may not have specified)... it just seems odd.

Lines 352 and 379: data are plural (data "were"...)

Line 609: shouldn't it be either "elevations have" or "elevation has" been reported?

Author Response

Reviewer 1:
This is a remarkably well-written submission. I found it to be both incredibly detailed and cautiously balanced. I suspect that it will be of tremendous use to the CF Care community. 

Author Response: We thank you for your very kind feedback regarding our review and its utility in the CF Care community.

I have one global comment and then a few minor suggestions.

The global comment is that I think the authors might want to consider the distinction between when someone is a "patient with CF" as opposed to when he or she is a "person with CF", as this has become something of a cause in the CF community.  When someone is admitted to the hospital they are clearly a patient, but is everyone out in the world taking a chronic medication or with a chronic condition always a patient?  Its not common to refer to people on statins as cardiac patients or pregnant women as obstetric patients, although they are presumably under the care of a physician... most of us are.   Perhaps the submission could be read in this context to identify where "patients" can become "people" (e.g., line 781: "As a growing number of patients with CF gain access to CFTR modulators").

Author Response: We thank you for identifying this important distinction and have since reviewed the manuscript to replace ‘patients’ with ‘people’, ‘individuals’, or an alternative suitable word in several places where appropriate.

Minor points:

Lines 287-288. The clause "In 1 study that specified,..." seems awkward. I think I understand the point being made (that other studies may not have specified)... it just seems odd.

Author Response: We agree that the sentence was awkwardly worded and have since revised it to: “One study specified that transaminases were over 6 times the upper limit of normal when discontinued.” [Line 312-313]

Lines 352 and 379: data are plural (data "were"...)

Author Response: We appreciate you identifying this grammatical error and have since revised accordingly.

Line 609: shouldn't it be either "elevations have" or "elevation has" been reported?

Author Response: We thank you for identifying this grammatical error and have since revised to “…elevations have…” [Line 657]

Reviewer 2 Report

In a manuscript by Renée Dagenais et al, the authors examined the effects of CFTR modulators therapy and adverse events associated real-world reported. The authors did huge amount of work searching online databases (MEDILNE, EMBASE, CINAHL and Web of Science) from 2012 to 2020. The review is well-written which satisfactorily describes the adverse events of the 4 drug combinations: Kalydeco, Orkambi, Symdeco and Trikafta. This highlights the need to share the real-world experiences with patients on CFTR modulators drugs to understand the long-term adverse events.

Minors:

Since the review is talking about the CFTR modulators, may be the authors could add a short paragraph talking about basic science of CFTR modulators and mutations.

Page 1 line 36: “resulting in dysfunction of the CFTR protein”. Since there are 6 classes of CFTR mutations, maybe it is better to say “resulting in alteration of CFTR protein synthesis, processing or function.

Page 2 line 58: please include the mutation after “gating and conductance”, for example (i.e. G551D).

Page 2 line 62: what does mean ³1 G551D? If you mean “at least G551D on one allele please mention “heterozygous”.

Tables: how do the authors decide if the final rating are Good or Fair? Please specify.

Author Response

Reviewer 2:
In a manuscript by Renée Dagenais et al, the authors examined the effects of CFTR modulators therapy and adverse events associated real-world reported. The authors did huge amount of work searching online databases (MEDILNE, EMBASE, CINAHL and Web of Science) from 2012 to 2020. The review is well-written which satisfactorily describes the adverse events of the 4 drug combinations: Kalydeco, Orkambi, Symdeco and Trikafta. This highlights the need to share the real-world experiences with patients on CFTR modulators drugs to understand the long-term adverse events.

Author Response: We thank you for your kind words regarding our manuscript.

Minors:

Since the review is talking about the CFTR modulators, may be the authors could add a short paragraph talking about basic science of CFTR modulators and mutations.

Author Response: We thank you for this suggestion. In the Introduction, we briefly touch upon the function of the CFTR channel in Paragraph 1 and the mechanism of how correctors and potentiators affect the CFTR channel in Paragraph 2. We are hesitant to delve into detail further due to the complex nature of this topic and in an attempt to maintain brevity, as the manuscript is already quite extensive.

We appreciate the suggestion and hope that not expanding beyond what we have already described does not detract from the quality of our manuscript.

Page 1 line 36: “resulting in dysfunction of the CFTR protein”. Since there are 6 classes of CFTR mutations, maybe it is better to say “resulting in alteration of CFTR protein synthesis, processing or function.

Author Response: We thank you for providing this helpful feedback. We agree with the suggested wording and have updated the sentence accordingly.  [Line 37-38]

Page 2 line 58: please include the mutation after “gating and conductance”, for example (i.e. G551D).

Author Response: We thank you for this feedback and have since included G551D and R117H as example mutations. [Line 63]